# Stage-specific control of oligodendrocyte survival and morphogenesis by TDP-43

Dongeun Heo[1], Jonathan P Ling[1,2], Gian C Molina-Castro[1], Abraham J Langseth[1], Ari Waisman[3], Klaus-Armin Nave[4], Wiebke Möbius[4,5,6], Phil C Wong[2], Dwight E Bergles[1,7]*

[1]The Solomon H. Snyder Department of Neuroscience, Johns Hopkins University School of Medicine, Baltimore, United States; [2]The Solomon H. Snyder Department of Neuroscience, Johns Hopkins University School of Medicine, Baltimore, United States; [3]Institute for Molecular Medicine, University Medical Center of the Johannes Gutenberg University, Mainz, Germany; [4]Department of Neurogenetics, Max Planck Institute of Experimental Medicine, Göttingen, Germany; [5]Cluster of Excellence "Multiscale Bioimaging: from Molecular Machines to Networks of Excitable Cells" (MBExC), University of Göttingen, Göttingen, Germany; [6]Electron Microscopy Core Unit, Max-Planck-Institute of Experimental Medicine, Göttingen, Germany; [7]Kavli Neuroscience Discovery Institute, Johns Hopkins University, Baltimore, United States

**Abstract** Generation of oligodendrocytes in the adult brain enables both adaptive changes in neural circuits and regeneration of myelin sheaths destroyed by injury, disease, and normal aging. This transformation of oligodendrocyte precursor cells (OPCs) into myelinating oligodendrocytes requires processing of distinct mRNAs at different stages of cell maturation. Although mislocalization and aggregation of the RNA-binding protein, TDP-43, occur in both neurons and glia in neurodegenerative diseases, the consequences of TDP-43 loss within different stages of the oligodendrocyte lineage are not well understood. By performing stage-specific genetic inactivation of *Tardbp* in vivo, we show that oligodendrocyte lineage cells are differentially sensitive to loss of TDP-43. While OPCs depend on TDP-43 for survival, with conditional deletion resulting in cascading cell loss followed by rapid regeneration to restore their density, oligodendrocytes become less sensitive to TDP-43 depletion as they mature. Deletion of TDP-43 early in the maturation process led to eventual oligodendrocyte degeneration, seizures, and premature lethality, while oligodendrocytes that experienced late deletion survived and mice exhibited a normal lifespan. At both stages, TDP-43-deficient oligodendrocytes formed fewer and thinner myelin sheaths and extended new processes that inappropriately wrapped neuronal somata and blood vessels. Transcriptional analysis revealed that in the absence of TDP-43, key proteins involved in oligodendrocyte maturation and myelination were misspliced, leading to aberrant incorporation of cryptic exons. Inducible deletion of TDP-43 from oligodendrocytes in the adult central nervous system (CNS) induced the same progressive morphological changes and mice acquired profound hindlimb weakness, suggesting that loss of TDP-43 function in oligodendrocytes may contribute to neuronal dysfunction in neurodegenerative disease.

*For correspondence: dbergles@jhmi.edu

## Editor's evaluation

Heo et al., 2021 investigated the role of the DNA/RNA-binding protein TDP-43 on oligodendrocyte maturation and showed that deletion of this protein induces deleterious effects on oligodendrocyte function in transgenic mouse models. These findings are exciting and support TDP-43 as a key regulator of oligodendrocyte function which may go awry in neurodegenerative disorders.

## Introduction

The distribution of myelin in the adult brain is not static, as new oligodendrocytes continue to be generated in response to changes in experience, such as enhanced sensory input and intense motor learning (*Hughes et al., 2018*; *Xiao et al., 2016*). As a result, oligodendrocyte lineage cells exist in a developmental continuum throughout life, consisting of oligodendrocyte precursor cells (OPCs) that maintain their density through proliferation (*Hughes et al., 2013*), and cells in the process of maturing from premyelinating to myelinating oligodendrocytes. This ongoing oligodendrogenesis is necessary to prolong the period of circuit maturation (*Benamer et al., 2020*), enable adaptive changes in myelination (*Gibson et al., 2014*), and restore myelin lost through aging or demyelinating injuries (*Orthmann-Murphy et al., 2020*; *Bacmeister et al., 2020*). The transition from progenitor to myelin-forming oligodendrocyte requires profound transcriptional changes, in which myelin-associated gene networks are activated and mRNAs are transported to distal processes to direct sites of axon ensheathment and membrane compaction (*Müller et al., 2013*). These transcriptional and post-transcriptional processes are orchestrated by RNA-binding proteins. However, the factors that coordinate key developmental transitions within the oligodendrocyte lineage to enable OPC homeostasis and myelin dynamics are not well understood.

TDP-43 (TAR DNA-binding protein 43 kDa) is a DNA/RNA-binding protein that serves as a master regulator of RNA metabolism, controlling transcriptional integrity, post-transcriptional stability, and localization of mRNAs (*Warraich et al., 2010*). Within the nucleus, TDP-43 prevents incorporation of cryptic exons into mature mRNA by binding to UG-rich motifs within intronic sequences (*Ling et al., 2015*). Loss of TDP-43 in neurons results in missplicing of pre-mRNAs, resulting in gain and loss of function alterations in the landscape of expressed proteins that influence their structure, function, and longevity (*Ling et al., 2015*; *Donde et al., 2019*; *Ma et al., 2021*; *Brown et al., 2021*). In many neurodegenerative diseases, including frontotemporal dementia, Alzheimer's disease (AD), amyotropic lateral sclerosis (ALS), and multiple sclerosis (MS), cytoplasmic mislocalization and aberrant aggregation of TDP-43 have been observed in affected regions of the CNS (*Neumann et al., 2006*; *Ling et al., 2015*; *Sun et al., 2017*; *Masaki et al., 2020*; *Salapa et al., 2020*). Moreover, mutations in *TARDBP* are responsible for both familial and sporadic forms of ALS (*Kapeli et al., 2017*; *Ince et al., 2011*), suggesting that altered mRNA processing by TDP-43 may contribute to cellular pathology and functional impairment in both sporadic and inherited forms of neurodegenerative disease. Although neurons have been the primary focus of research into TDP-43-associated pathologies, nuclear exclusion, and cytoplasmic aggregation of TDP-43 also occur in glial cells in neurodegenerative diseases, such as MS (*Neumann et al., 2006*; *Masaki et al., 2020*; *Rohan et al., 2014*), suggesting that glial cell dysfunction could impair both OPC homeostasis and remyelination. However, much less is known about the role of TDP-43 in mRNA metabolism or the impact of its loss within oligodendroglia at distinct stages of maturation.

Genetic deletion of TDP-43 in mice results in embryonic lethality due to impaired expansion of inner cell mass during early embryogenesis (*Sephton et al., 2010*), suggesting that it is essential for cell survival. However, selective deletion experiments indicate that the effects of TDP-43 are cell type specific and diverse (*Jeong et al., 2017*), reflecting differences in transcriptional state. In neurons, conditional inactivation of TDP-43 induces impairment in neuronal function and progressive neurodegeneration (*Donde et al., 2019*; *Iguchi et al., 2013*; *Yang et al., 2014*; *Wu et al., 2019*). Conditional knockout of *Tardbp* in microglia enhanced expression of NF-κB and NLRP3, leading to a more activated phenotype characterized by enhanced phagocytosis (*Zhao et al., 2015*; *Paolicelli et al., 2017*), while loss of TDP-43 in astrocytes promoted their transition to a reactive A1 phenotype associated with neuronal injury (*Peng et al., 2020*; *Walker et al., 2014*). In Schwann cells, genetic deletion of *Tardbp* did not affect their survival or myelinic potential, but impaired formation and maintenance of paranodal junctions (*Chang et al., 2021*), while deletion from both newly generated oligodendrocytes and Schwann cells induced necroptosis of oligodendrocytes and early mortality (*Wang et al., 2018*). These results indicate that TDP-43 regulates transcriptional integrity of different genes based on cellular context, highlighting the need to define TDP-43 functions within different cellular states. As oligodendroglia consist of both proliferating progenitors and cells in distinct states of maturation, their unique transcriptional profiles may render them differentially sensitive to loss of TDP-43.

Here, we used in vivo targeted genetic manipulations of the oligodendrocyte lineage to define the consequences of TDP-43 loss of function in the mouse brain. OPCs were the most vulnerable to this

manipulation, as selective deletion of TDP-43 from OPCs resulted in their rapid death, progressing from the corpus callosum to the overlying cortical gray matter. However, this progenitor depletion was transient, as homeostatic proliferation of OPCs that retained TDP-43 expression led to rapid repopulation and restoration of their normal density. Unexpectedly, the consequences of deletion of TDP-43 in oligodendrocytes were dependent on their stage of maturation. Early deletion resulted in the progressive degeneration of mature oligodendrocytes, leading to seizures and premature lethality. In contrast, late deletion largely spared oligodendrocytes, and mice survived without seizures, indicating that TDP-43 may be dispensable within mature oligodendrocytes. Strikingly, both early and late manipulation of TDP-43 in oligodendrocytes led to profound morphological changes, accompanied by thinner and fewer myelin sheaths, as well as progressive, inappropriate wrapping of neuronal somata and blood vessels. Conditional removal of TDP-43 from oligodendrocytes in the adult CNS induced similar morphological changes and was associated with profound hindlimb weakness. Differential transcriptional analysis of oligodendrocytes revealed that loss of TDP-43 led to missplicing of RNAs encoding key genes involved in oligodendrocyte morphogenesis and myelination, changes that may contribute to abnormal neuronal activity and neurodegeneration in diverse neurological diseases.

## Results

### Selective deletion of TDP-43 from OPCs induces their progressive loss

Oligodendrocytes are derived from an abundant, widely distributed population of lineage restricted progenitors, termed OPCs, that maintain their density through local homeostatic proliferation (*Hughes et al., 2013*). The dynamic nature of these cells may render them particularly susceptible to environmental and genetic perturbations. To determine how loss of TDP-43 affects OPCs in vivo, we selectively deleted *Tardbp* from OPCs in young adult *Pdgfra-CreER;Tardbp^{fl/fl};RCE* (PDGFRα-TDP43 cKO) mice, by administering tamoxifen at P90 and then collecting brain tissue 2 weeks later (P90 + 14) (*Figure 1A*). Inclusion of the RCE transgene (*Rosa-CAG-LSL-EGFP*) allows the visualization of the morphology of Cre-recombined cells (*Sousa et al., 2009*). Immunolabeling for EGFP and NG2, a proteoglycan highly expressed by OPCs, revealed that this manipulation induced recombination in most OPCs (EGFP+NG2+/NG2+ cells: 94.75 ± 0.75%, $n$ = 4) (*Figure 1B*), resulting in near complete depletion of OPCs from the corpus callosum, while those in the overlying cortical gray matter were preserved (*Figure 1C,E*; *Figure 1D*, panel i) (OPCs/mm³: Corpus callosum 14 dpi, Control = 6910 ± 201; cKO = 987 ± 201, $n$ = 3, p < 0.001, one-way ANOVA with Tukey's multiple comparisons test). However, when the interval between tamoxifen administration and sacrifice was increased to 30 days (P90 + 30), OPCs in gray matter were also depleted (*Figure 1C,E*; *Figure 1D*, panel ii) (OPCs/mm³: Cortex 30 dpi, Control = 5247 ± 236; cKO = 811 ± 187, $n$ = 3, p < 0.001, one-way ANOVA with Tukey's multiple comparisons test). Remarkably, OPCs in white matter had repopulated by this time, and highly polarized OPCs were visible migrating out of the corpus callosum into the gray matter (*Figure 1C*, red arrowhead; *Figure 1D*, panel iii; *Figure 1E*). When the interval between tamoxifen administration and sacrifice was increased by another 30 days (P90 + 60), OPCs had regained their normal density and tiled distribution throughout gray and white matter (*Figure 1C,E*). As expected, this repopulation was associated with enhanced proliferation of OPCs, visible by increased Ki-67 immunoreactivity in PDGFRα-TDP43 cKO (*Figure 1F*; *Figure 1—figure supplement 1A*). Despite this extensive death of OPCs, there were no signs of astrogliosis, as assessed by glial fibrillary acidic protein (GFAP) immunoreactivity (*Figure 1—figure supplement 1B*), in accordance with the tolerance to normal turnover of these cells (*Hughes et al., 2013*). This lack of glial reactivity in response to widespread loss of OPCs was also described in a previous OPC ablation study using diphtheria toxin (*Birey et al., 2015*), suggesting that OPC death does not induce reactive changes in astrocytes or microglia. Deletion of *Tardbp* in P180 mice also resulted in a robust loss of OPCs in the corpus callosum within 14 days (P180 + 14) (*Figure 1—figure supplement 1C*), indicating that the dependence on TDP-43 extends into the adult CNS. Together, these results show that TDP-43 is required for the survival and homeostasis of oligodendrocyte progenitors.

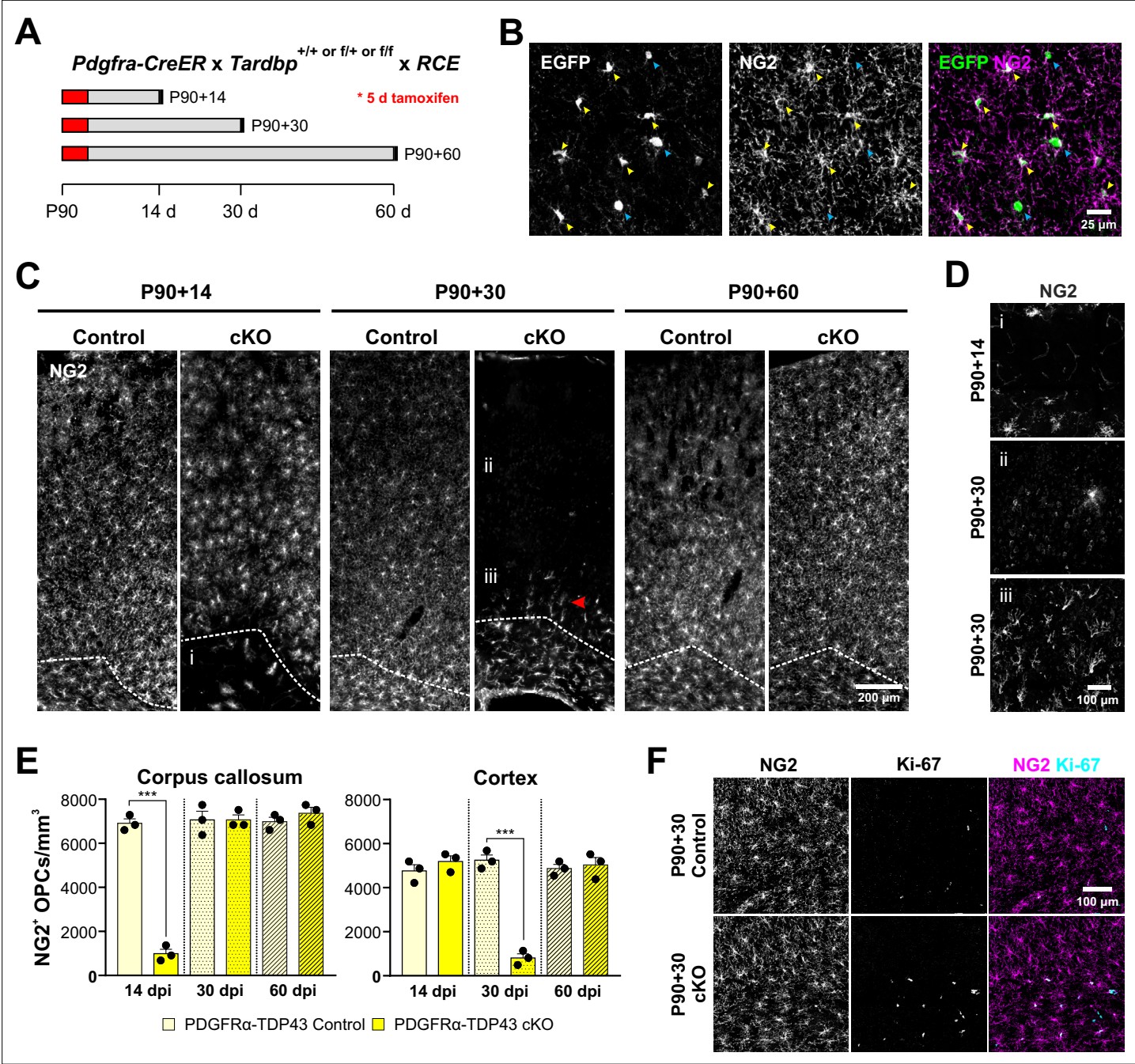

**Figure 1.** Oligodendrocyte precursor cells (OPCs) require TDP-43 for survival. (**A**) Schematics of CreER activation. At P90, tamoxifen was administered for five consecutive days. Tissue samples were collected 14, 30, and 60 days after the last day of tamoxifen injection. (**B**) *Pdgfra-CreER;RCE* allows labeling of oligodendrocyte lineage cells. Yellow arrowheads indicate EGFP+NG2+ OPCs whereas blue arrowheads indicate EGFP+NG2– oligodendrocytes. (**C**) NG2 staining shows that there is a regional progression in OPC degeneration from white matter to gray matter. The dotted line separates the corpus callosum from cortex. i, ii, and iii regions are zoomed in in D. (**D**) Region i shows the corpus callosum with a near complete loss of OPCs at P90 + 14. ii shows loss of OPCs in the cortex at P90 + 30. iii shows highly polarized OPCs migrating from the corpus callosum up to cortex. (**E**) Quantification of NG2+ OPC density in corpus callosum and cortex at 14, 30, and 60 days post injection of tamoxifen (dpi). Statistical significance was determined using one-way ANOVA with Tukey's multiple comparisons test ($n = 4$, ***p value < 0.001, n.s. p value >0.05). (**F**) Ki-67+NG2 + proliferating OPCs increase in number at P90 + 60 in the corpus callosum of PDGFRα-TDP43 KO.

The online version of this article includes the following figure supplement(s) for figure 1:

**Figure supplement 1.** PDGFRα-TDP43 cKO shows a regional difference in degeneration of oligodendrocyte precursor cells (OPCs).

# Deletion of TDP-43 from premyelinating oligodendrocytes impairs oligodendrocyte survival

OPCs undergo a profound series of structural and physiological changes as they differentiate into myelinating oligodendrocytes, a transition enabled by activation of distinct gene regulatory networks expressed at different stages of their maturation (*Xu et al., 2020*; *He et al., 2017*). To determine if postmitotic oligodendrocytes are similarly dependent on TDP-43, we deleted *Tardbp* from these cells using two constitutive Cre lines which differ in their timing of expression during oligodendrocyte maturation: *Mog*[iCre], which has been used extensively to manipulate mature oligodendrocytes (*Buch et al., 2005*; *Larson et al., 2018*; *Figure 2—figure supplement 1*) and *Mobp-iCre*, a novel BAC transgenic line that we developed (see Materials and methods) (*Figure 2—figure supplement 2*). Single-cell RNA-Seq results indicate that *Mobp* expression is robust at the premyelinating oligodendrocyte stage, whereas expression of *Mog* does not increase substantially until the myelin-forming stage (*Marques et al., 2016*; *Hrvatin et al., 2018*; *Figure 2—figure supplement 3*). Gene expression analysis of oligodendroglia at these stages revealed that many mRNAs are differentially expressed, including *Gadd45a*, which participates in DNA methylation, and *Opalin* (Tmem10) that has been shown to promote oligodendrocyte differentiation (*Barreto et al., 2007*; *de Faria et al., 2019*; *Figure 2—figure supplement 4*), suggesting that the consequences of TDP-43 may be distinct.

Abolishing TDP-43 expression at these different stages resulted in very different outcomes. Mice in which TDP-43 was constitutively removed from premyelinating oligodendrocytes (*Mobp-iCre;Tardbp*[fl/fl]*;RCE*) (Mobp-TDP43 KO) began to exhibit spontaneous seizures at ~P110 (*Figure 2B*, *red arrowhead*; *Video 1*) and 100% died prematurely by P180 (*Figure 2B*, *dark orange line*), similar to the phenotype exhibited by *Cnp*[Cre]*;Tardbp*[fl/fl] mice (*Wang et al., 2018*), in which recombination occurs in OPCs, early postmitotic oligodendrocytes, and Schwann cells in the PNS (*Lappe-Siefke et al., 2003*; *Chang et al., 2021*; *Tognatta et al., 2017*). In contrast, mice in which TDP-43 was deleted from mature oligodendrocytes (*Mog*[iCre]*;Tardbp*[fl/fl]*;RCE*) (Mog-TDP43 KO) did not exhibit spontaneous seizures or other gross behavioral abnormalities and survived as well as controls (*Figure 2B*, *blue lines*).

In contrast to the extensive demyelination and oligodendrocyte necroptosis observed in *Cnp*[Cre]*;Tardbp*[fl/fl] mice (*Wang et al., 2018*), there was no change in the density of ASPA+ mature oligodendrocytes in the motor cortex of Mobp-TDP43 KO or Mog-TDP43 KO mice at P30, P90, or P180 (end stage) (*Figure 2C,D*). However, degenerating oligodendrocytes can be rapidly replaced through differentiation of OPCs (*Kang et al., 2013*), obscuring extensive cell loss. As remaining OPCs proliferate to maintain their density, higher rates of OPC proliferation are associated with oligodendrocyte degeneration (*Kang et al., 2013*). In accordance with the different behavioral phenotypes of the Mobp-TDP43 KO and Mog-TDP43 KO mice, OPCs also exhibited strikingly different behaviors in these animals. Early deletion of TDP-43 in premyelinating oligodendrocytes (Mobp-TDP43 KO) was associated with a dramatic increase in Ki-67+ OPCs at P90 (*Figure 2E,F*) (Ki-67+ OPCs/mm$^3$: Mobp-TDP43 Control = 71 ± 45; Mobp-TDP43 KO = 941 ± 121, $n = 4$, $p < 0.001$, one-way ANOVA with Tukey's multiple comparisons test), whereas deletion of TDP-43 from mature oligodendrocytes (Mog-TDP43 KO) was not associated with a significant change in OPC proliferation (*Figure 2E,F*) (Ki-67+ OPCs: Mog-TDP43 Control = 24 ± 24; Mog-TDP43 KO = 212 ± 89, $n = 4$, $p > 0.05$, one-way ANOVA with Tukey's multiple comparisons test). Moreover, there was an increase in the number of newly formed oligodendrocytes in Mobp-TDP43 KO, as assessed by the emergence of cells that expressed lncOL1 (*Figure 2—figure supplement 4A*), a long noncoding RNA that is transiently expressed in premyelinating oligodendrocytes (*He et al., 2017*), suggesting that there is an increased turnover of oligodendrocytes. Astrogliosis (visible through the increase in GFAP immunoreactivity) was also prominent in Mobp-TDP43 KO mice (*Figure 2—figure supplement 4B, C*), further supporting the conclusion that early deletion of TDP-43 leads to degeneration and subsequent regeneration of oligodendrocytes. Together, these results indicate that there is a critical period for TDP-43 function during oligodendrocyte maturation, with deletion at an early stage leading to progressive oligodendrocyte degeneration, seizures, and premature death, consequences that are avoided when oligodendrocytes mature beyond this stage.

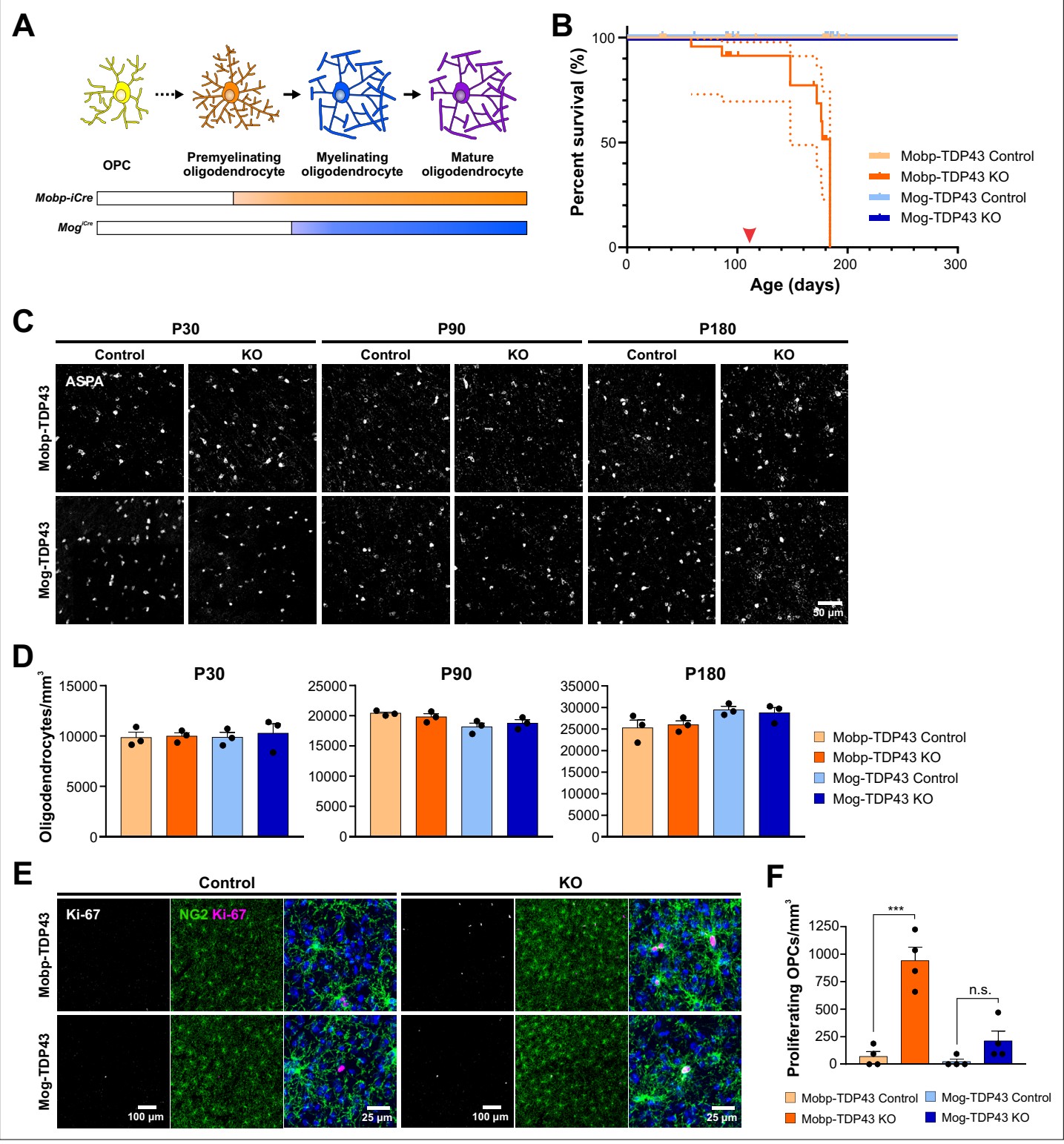

**Figure 2.** Early deletion of TDP-43 in premyelinating oligodendrocytes leads to seizure development, premature death, and increased oligodendrocyte turnover. (**A**) Diagram of oligodendrocyte development showing *Mobp-iCre* induces *Tardbp* deletion at the premyelinating oligodendrocyte stage whereas *Mog-iCre* targets myelinating oligodendrocytes. (**B**) Kaplan–Meier survival curve for Mobp-TDP43 (orange) and Mog-TDP43 (blue) mouse lines. Dotted line indicates the 95% confidence interval. Log-rank (Mantel–Cox) statistical test showed significance at p value <0.0001 (*n* = 58, 33, 40, and 32). Red arrowhead points to ~P110 when the animals first exhibit spontaneous seizures. (**C**) ASPA+ oligodendrocyte staining in the motor cortex of Mobp-TDP43 and MOG-TDP43 mouse lines at P30, P90, and P180. (**D**) Quantification of ASPA+ oligodendrocyte density shows that there is no

*Figure 2 continued on next page*

*Figure 2 continued*

statistical difference between the samples at any given timepoint (one-way ANOVA with Tukey's multiple comparisons test, $n = 3$, n.s. p value >0.05). (**E**) Immunostaining for Ki-67 and NG2 to identify proliferating oligodendrocyte precursor cells (OPCs) in Mobp-TDP43 and Mog-TDP43 at P90. (**F**) Quantification of Ki-67+NG2+ proliferating OPC density shows that it is statistically significantly increased in Mobp-TDP43 KO whereas Mog-TDP43 KO shows a trend toward an increased density (one-way ANOVA with Tukey's multiple comparisons test, $n = 4$, ***p value <0.001, n.s. p value = 0.3776).

The online version of this article includes the following figure supplement(s) for figure 2:

**Figure supplement 1.** Quantification of *Mog-iCre* recombination efficiency.

**Figure supplement 2.** Characterization of *Mobp-iCre* and *Mobp-iCreER^{T2}* mouse lines.

**Figure supplement 3.** Mobp-TDP43 and Mog-TDP43 mouse lines exhibit differences in response to loss of TDP-43.

**Figure supplement 4.** Differential gene expression between premyelinating (orange) and myelinating (blue) oligodendrocytes (OL) (reproduced from the raw single-cell RNA-Seq dataset in *Hrvatin et al., 2018*).

**Figure supplement 5.** Mobp-TDP43 KO animals exhibit a higher oligodendrocyte turnover rate and increased astrogliosis.

## TDP-43 loss in oligodendrocytes results in thinner myelin, fewer myelinated axons, and aberrant nodes of Ranvier

In both Mobp-TDP43 KO and Mog-TDP43 KO mice, oligodendrocytes were able to mature and form myelin sheaths, global myelin tracts within the brain were established (*Figure 3—figure supplement 1A*), and there was no evidence of astrogliosis early in development (P30) (*Figure 3—figure supplement 2*). To determine if oligodendrocytes in these mice were capable of forming normal myelin at the ultrastructural level, we performed transmission electron microscopy (TEM) of the corpus callosum (*Figure 3—figure supplement 1A*, *red boxes*). Myelinated axons in Mobp-TDP43 KO and Mog-TDP43 KO mice had significantly higher *G* ratios than controls at both P90 and P180, with thinner sheaths formed (*Figure 3A–D*). Although axons with diameters less than 0.6 μm were not as severely affected at P90, their myelin was also markedly thinner in P140–180 mice (*Figure 3—figure supplement 1B*). In addition to thinner myelin sheaths, the number of myelinated axons was also reduced in both oligodendrocyte TDP-43 knockouts (*Figure 3E,F*). Morphological reconstructions of individual oligodendrocytes and their associated myelin sheaths revealed that Mog-TDP43 KO oligodendrocytes tended to exhibit less myelin basic protein (MBP) immunoreactivity per process (revealed by EGFP labeling), indicative of decreased myelin production (*Figure 3—figure supplement 3*). These results indicate that despite achieving a mature myelinating stage, the total myelin output of TDP-43-deficient oligodendrocytes is ultimately lower (summarized in *Figure 3—figure supplement 4*).

In peripheral nerves, loss of TDP-43 from Schwann cells results in complete loss of paranodal junctions and a 50% decrease in conduction velocity (*Chang et al., 2021*). Immunocytochemical analysis revealed that TDP-43-null oligodendrocytes also frequently exhibited abnormal nodes of Ranvier (NOR). While normal NOR exhibit focal localization of βIV spectrin at the node flanked by Caspr within the paranodes (*Figure 3—figure supplement 3A*, *yellow arrowheads*), in mice with TDP-43 KO oligodendrocytes, nodes often consisted of Caspr triads and βIV-spectrin doublets (*Figure 3—figure supplement 5A*, *red arrowheads*). EM analysis revealed that some axons exhibited double myelination (*Figure 3—figure supplement 1C*), which may contribute to these macroscopic changes in organization of node-associated proteins (*Figure 3—figure supplement 5B*), as well as unusual accumulation of mitochondria and vacuoles in the inner tongue.

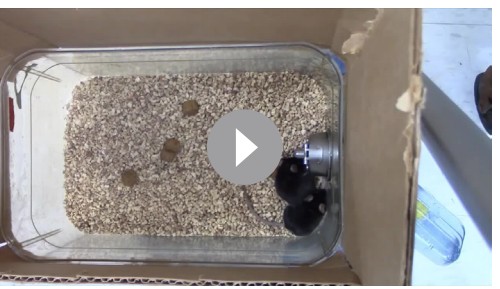

**Video 1.** Video of Mobp-TDP43 Control and KO mice where Mobp-TDP43 KO mice undergo spontaneous seizures and reduced threshold for seizures.
https://elifesciences.org/articles/75230/figures#video1

## Oligodendrocytes without TDP-43 undergo profound morphological changes

To define the progressive structural changes exhibited by TDP-43-deficient oligodendrocytes, we examined their morphologies in Mobp-TDP43 KO and Mog-TDP43 KO mice using EGFP (expressed from the conditional RCE transgene) (*Sousa et al., 2009*). Histological analysis of

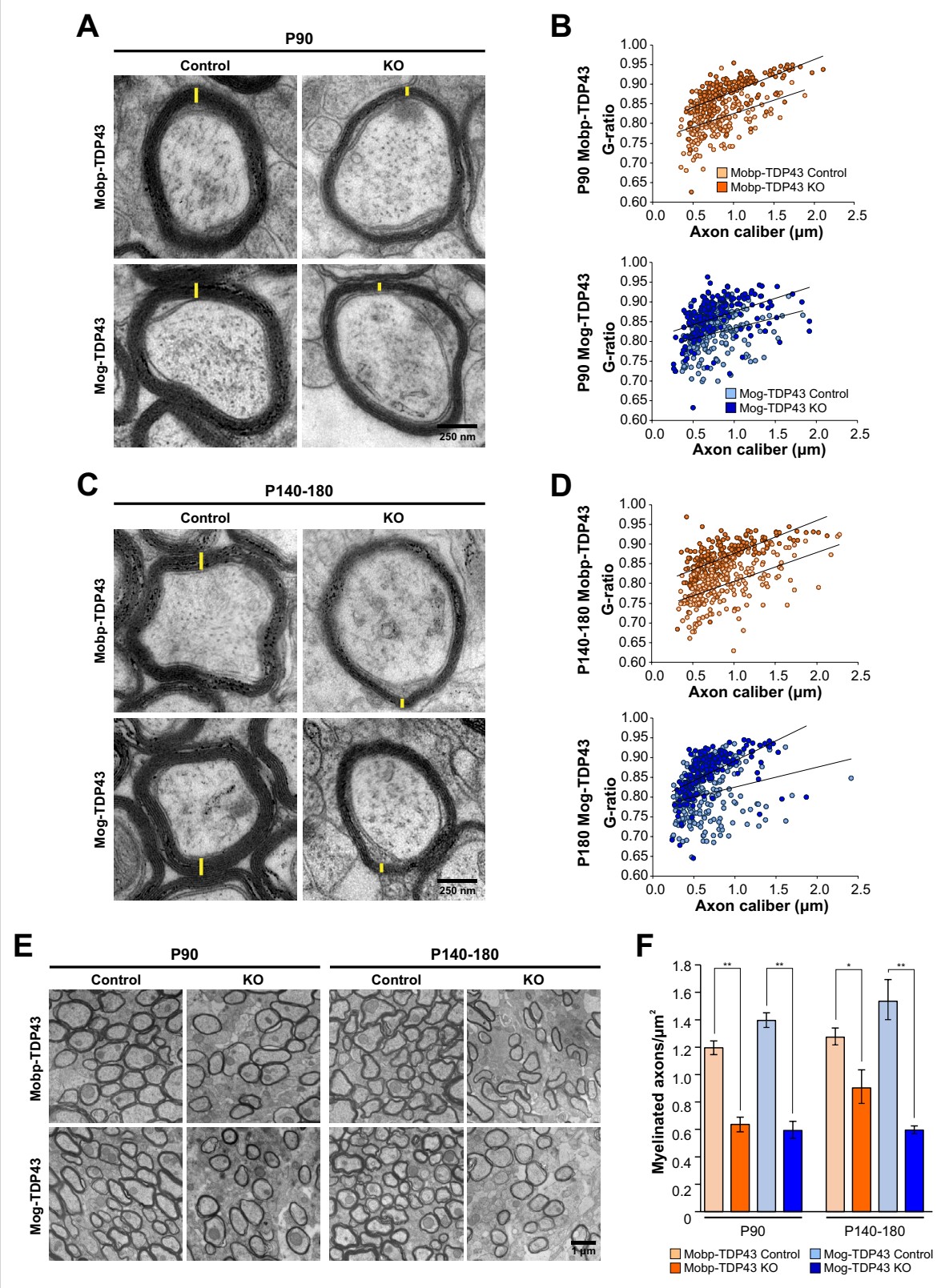

**Figure 3.** Loss of TDP-43 oligodendrocytes leads to fewer and thinner myelin sheaths. (**A**) Representative transmission electron micrographs of individual axon cross-sections from Mobp-TDP43 and Mog-TDP43 mouse lines at P90 show decreased myelin sheath thickness (yellow) in the KO. Scale bar = 250 nm. (**B**) *G* ratios of individual myelinated axons as a function of axon caliber (µm) in the corpus callosum of Mobp-TDP43 (orange) and Mog-TDP43 (blue) mouse lines at P90. *n* = 3–4/genotype. The solid lines show the lines of best fit with a linear function. (**C**) A for P140–180 timepoint. (**D**) B

*Figure 3 continued on next page*

*Figure 3 continued*

for P140–180 timepoint. (**E**) Representative electron micrographs of Mobp-TDP43 and Mog-TDP43 mouse lines at P90 and P140–180 show a decreased density of myelinated axons in the KO. Scale bar = 1 μm. (**F**) Quantification of myelinated axon density in Mobp-TDP43 (orange) and Mog-TDP43 (blue) mouse lines at P90 and P140–180. Statistical significance was determined using unpaired, two-tailed Student's *t*-test (**p value <0.01, *p value <0.05, n.s. p value >0.05).

The online version of this article includes the following figure supplement(s) for figure 3:

**Figure supplement 1.** Both Mobp-TDP43 KO and Mog-TDP43 KO brains show significantly increased g-ratio and aberrant myelination.

**Figure supplement 2.** Immunostaining for GFP, GFAP, and myelin basic protein (MBP) at P30 in Mobp-TDP43 and Mog-TDP43 mouse lines shows that oligodendrocytes and myelin are formed without astrogliosis.

**Figure supplement 3.** Morphological reconstruction of individual oligodendrocytes and their myelin sheaths.

**Figure supplement 4.** Diagram of Control (green) and KO (red) oligodendrocytes, where KO oligodendrocytes are making fewer and thinner myelin sheaths.

**Figure supplement 5.** TDP-43 loss in oligodendrocytes leads to abnormal nodal structures.

motor cortex in these mice revealed that oligodendrocyte density gradually increased from P30 to P180 in both control and mutant (KO) mice (***Figure 2D***); however, while EGFP was largely restricted to oligodendrocyte somata in controls, due to the limited cytoplasm in myelin sheaths, mice with TDP-43-deficient oligodendrocytes exhibited a dramatic increase in EGFP fluorescence in the surrounding parenchyma at P90 and P180 (***Figure 4—figure supplement 1***). This morphological change in oligodendrocytes was observed throughout the brain, as well as the optic nerve and spinal cord (***Figure 4—figure supplement 2***), suggesting that this phenotype is fully penetrant in the CNS. Repetitive in vivo two photon imaging of individual oligodendrocytes in the motor cortex of Mobp-TDP43-RCE KO mice over 2 weeks revealed that these aberrant structures are highly dynamic (***Figure 4—figure supplement 3***; ***Video 2***), indicating that these normally highly stable cells exhibit enhanced growth when TDP-43 is absent. High-resolution imaging revealed that much of this additional EGFP fluorescence arose from thick cytoplasmic processes of oligodendrocytes that often ended in circular or cuff-like structures (***Figure 4A,B***). Immunostaining for NeuN revealed that circular structures arose from inappropriate wrapping of neuronal cell bodies (***Figure 4C,D***; ***Video 3***). These abnormal associations were not rare events and were visible in single coronal brain sections immunostained for NeuN and EGFP (***Figure 4—figure supplement 4***).

As the cuff-like structures are reminiscent of astrocyte endfeet (***Gundersen et al., 2014***), we labeled blood vessels with GS-Isolectin, revealing that EGFP+ oligodendrocyte processes also colocalized with these tubular structures (***Figure 4E***). To determine if these structures were capillaries, we used in vivo two photon imaging in living Mobp-TDP43-RCE KO mice to visualize both oligodendrocyte processes and capillaries after a retro-orbital injection of dextran–rhodamine. With 4D imaging, it was possible to follow the cytoplasmic processes from individual oligodendrocytes, revealing that they extended unbranched processes to ensheath rhodamine-labeled capillaries (***Figure 4F***). Histological analysis revealed that regions of capillaries surrounded by oligodendrocyte cuffs exhibited reduced aquaporin four immunoreactivity (***Figure 4—figure supplement 5A***), a protein restricted to astrocyte endfeet (***Gundersen et al., 2014***), suggesting that these oligodendrocyte wraps may displace or alter the physiology of astrocytes at capillaries. Remarkably, we also found that some TDP-43 KO oligodendrocytes exhibited immunoreactivity to Ki-67 (***Figure 4—figure supplement 5B***), suggesting that these cells are reverting to a less mature phenotype. Together, these results indicate that TDP-43 is required maintain the structure of

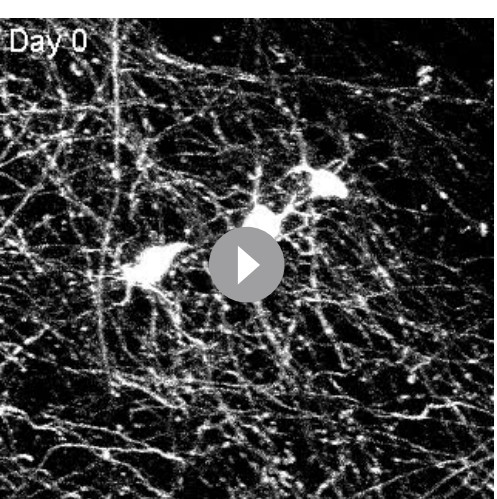

**Video 2.** Longitudinal in vivo imaging of Mobp-TDP43 KO oligodendrocytes.
https://elifesciences.org/articles/75230/figures#video2

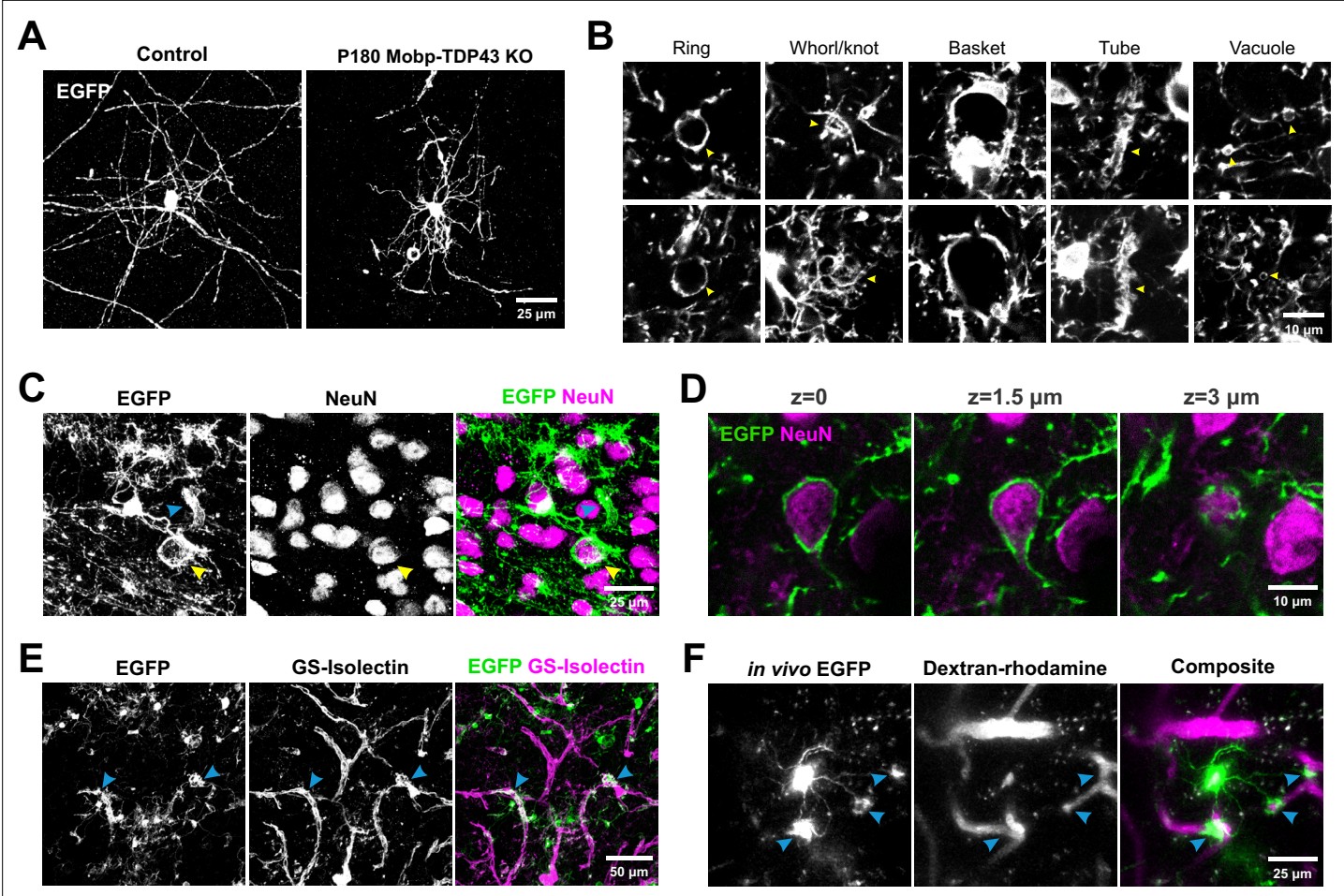

**Figure 4.** Oligodendrocytes without TDP-43 undergo aberrant morphological changes and exhibit aberrant wrapping of nonaxonal structures. (**A**) Representative images of single oligodendrocytes from Control (*Mobp-EGFP*) and Mobp-TDP43 KO at P180. Scale bar = 25 μm. (**B**) Examples of abnormal structures formed by TDP-43 KO oligodendrocytes. Yellow arrowheads point to abnormal structures for each category. Scale bar = 10 μm. (**C**) Immunostaining of GFP and NeuN in Mobp-TDP43-RCE KO shows a formation of basket-like EGFP+ structure that ensheaths NeuN+ neuronal cell body (yellow arrowhead). Blue arrowhead points to a tubing-like EGFP+ structure. Scale bar = 25 μm. (**D**) Optical serial sections representing 1.5 μm increments in the *z*-axis showing wrapping of a NeuN+ neuronal soma by EGFP+ oligodendrocyte processes in Mobp-TDP43-RCE KO at P180. Scale bar = 10 μm. (**E**) Immunostaining of GFP and GS-Isolectin in Mog-TDP43-RCE KO at P180 shows tight wrapping of the blood vessels by EGFP+ oligodendrocyte processes (blue arrowheads). Scale bar = 50 μm. (**F**) In vivo imaging of Mobp-TDP43-RCE KO mouse after retro-orbital injection of 70 kDa dextran–rhodamine shows wrapping of the capillaries by EGFP+TDP-43 KO oligodendrocyte. Blue arrows show sites of wrapping. Scale bar = 25 μm.

The online version of this article includes the following figure supplement(s) for figure 4:

**Figure supplement 1.** Oligodendrocytes undergo progressive morphological changes with loss of TDP-43.

**Figure supplement 2.** Abnormal oligodendrocyte morphological phenotype due to loss of TDP-43 is persistent throughout the CNS.

**Figure supplement 3.** Longitudinal two-photon in vivo imaging of Mobp-TDP43 KO oligodendrocytes across 14 days.

**Figure supplement 4.** Aberrant wrapping of non-axonal structures is prevalent in Mobp-TDP43 KO brains.

**Figure supplement 5.** Loss of TDP-43 in oligodendrocytes potentially contributes to the disruption of blood-brain barrier (BBB) and de-differentiation of oligodendrocytes.

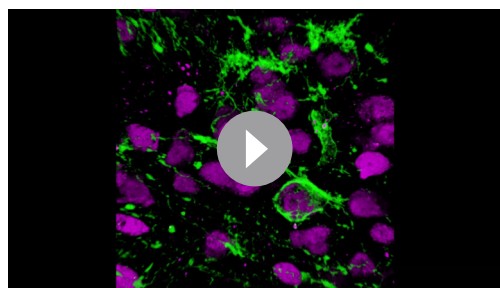

**Video 3.** 3D video of EGFP+ KO oligodendrocyte that wrap around NeuN+ neuronal somata.
https://elifesciences.org/articles/75230/figures#video3

oligodendrocytes and myelin throughout the CNS.

## Conditional deletion of TDP-43 from oligodendrocytes in the adult CNS induces abnormal morphological changes and motor discoordination

The results above indicate that constitutive deletion of TDP-43 from myelinating oligodendrocytes induces a profound morphological transformation, and when removed early in their maturation, results in oligodendrocyte degeneration, seizures, and death. Although oligodendrocytes continue to be generated in the adult CNS, most oligodendrocytes are produced in the first two postnatal months (*Bergles and Richardson, 2015*), indicating that these changes primarily manifest in developing tissue. To determine if loss of TDP-43 function in oligodendrocytes in the mature brain evokes comparable changes, we generated *Mobp-iCreER* mice, which allow selective Cre-dependent recombination in oligodendrocytes in the adult CNS (see Materials and methods; *Figure 2—figure supplement 2*). When tamoxifen was administered to P90 *Mobp-iCreER;Tardbp*^fl/fl^;*RCE* mice and analyzed 30 days later (P90 + 30), ASPA+ oligodendrocytes were present in the motor cortex with morphologies comparable to controls (*Mobp-iCreER;Tardbp*^+/+, fl/+^) (*Figure 5A,B*). However, when TDP-43 mutant mice were analyzed 90 days later (P90 + 90), the majority of oligodendrocytes exhibited hypertrophy, as well as cytoplasmic ensheathment of neuronal cell bodies and capillaries (*Figure 5B*, *yellow arrowhead*), indicating that TDP-43 is also required to maintain oligodendrocyte structure in the adult CNS. However, the behavioral consequences of TDP-43 deletion in the mature CNS were distinct, as these mice exhibited progressive motor problems, including limp tails, hindlimb clasping (*Figure 5C*, *Video 4*), and hindlimb weakness (*Figure 5D*), but not seizures or premature lethality. We utilized the clinical scoring method established for experimental autoimmune encephalomyelitis (EAE), a mouse model of MS, and found that Mobp-TDP43 cKO developed significant motor weakness within 90 days after Cre recombination (*Figure 5E*). Although this transformation of oligodendrocytes did not lead to degeneration of motor neurons (*Figure 5—figure supplement 1*), the spinal cord exhibited enhanced GFAP immunoreactivity (*Figure 5F,G*), indicative of widespread inflammation, which has been shown to impair motor function without loss of spinal motor neurons (*Wang et al., 2018*). Thus, adult-onset loss of TDP-43 function in oligodendrocytes, which may occur in disease and injury states (*Wiesner et al., 2018*; *Sun et al., 2017*), triggers phenotypes commonly observed in mouse models of ALS, Parkinson's disease (PD), and MS (*Wong and Martin, 2010*; *Lieu et al., 2013*; *Gharagozloo et al., 2021*).

## TDP-43 regulates splicing and represses cryptic exon incorporation in oligodendrocytes

TDP-43 is a highly conserved, multifunctional protein that regulates key aspects of RNA processing. Previous studies have shown that loss of nuclear TDP-43 function, through genetic deletion or disease-induced mislocalization, is sufficient to cause aberrant incorporation of intronic sequences (cryptic exons) into mature mRNA that can alter protein expression and function (*Ling et al., 2015*; *Sun et al., 2017*; *Donde et al., 2019*). Due to the unique complement of RNAs present in different cell types and differences in intron–exon boundaries between organisms, these changes are both cell type and species specific (*Jeong et al., 2017*). To define the molecular consequences of TDP-43 loss of function in oligodendrocytes, we used fluorescence-activated cell sorting (FACS) to selectively isolate oligodendrocytes from the cortex of P30 Mobp-TDP43-RCE KO and Mog-TDP43-RCE KO mice (and controls) prior to the onset of phenotypic changes (*Figure 4A*; *Figure 3—figure supplement 1*), and performed bulk mRNA sequencing (RNA-Seq) to determine gene expression changes (*Figure 6A*). In both mouse lines, hundreds of genes were differentially expressed, comprising 3–6% of the mouse genome (*Figure 6B*; *Figure 6—figure supplement 1*), consistent with the magnitude of changes

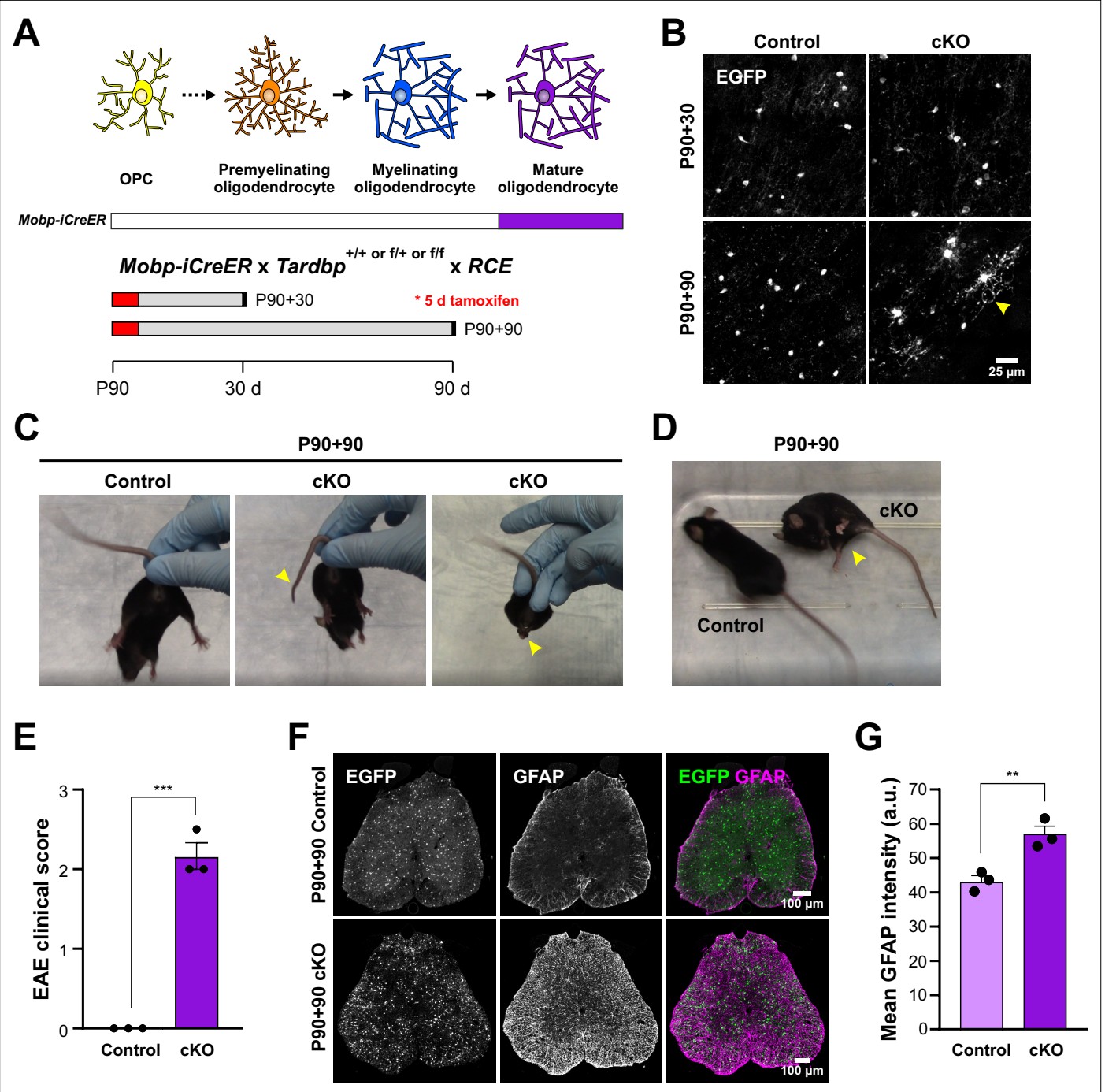

**Figure 5.** Adult loss of TDP-43 in oligodendrocytes leads to hindlimb weakness. (**A**) *Mobp-iCreER* allows *Tardbp* deletion at mature oligodendrocytes in the adult CNS. Schematics of CreER activation where tamoxifen is administered for five consecutive days at P90. Samples are collected for analyses 30 and 90 days after the last day of tamoxifen injection. (**B**) EGFP images from Mobp-TDP43-RCE Control and cKO at 30 and 90 days after *Tardbp* deletion. Yellow arrowhead indicates abnormal morphological changes in cKO oligodendrocytes at P90 +90. Scale bar = 25 µm. (**C**) Examples of limp tail and hindlimb clasping (yellow arrowheads) in Mobp-TDP43 cKO at P90 + 90 from *Video 4*. (**D**) Mobp-TDP43 cKO mice develop hindlimb paralysis at P90 + 90 (yellow arrowhead). (**E**) Experimental autoimmune encephalomyelitis (EAE) clinical score for Control and Mobp-TDP43 cKO mice. All Control mice at P90 + 90 exhibit no obvious changes in motor function whereas Mobp-TDP43 cKO mice develop limp tail and hindlimb weakness, which yield clinical scores between 2 and 2.5 (unpaired, two-tailed Student's *t*-test, ***p value = 0.0002, n = 3). (**F**) Immunostaining for GFP and GFAP in the lumbar spinal cords of Mobp-TDP43-RCE Control and cKO at P90 + 90 shows global astrogliosis indicated by increased immunoreactivity of GFAP. Scale bar = 100 µm. (**G**) Quantification of mean GFAP fluorescence intensity (arbitrary unit; a.u.) shows a statistically significant increase in the mean intensity of GFAP in the spinal cord of Mobp-TDP43 cKO (unpaired, two-tailed Student's *t*-test, **p value = 0.0096, n = 3).

*Figure 5 continued on next page*

*Figure 5 continued*

The online version of this article includes the following figure supplement(s) for figure 5:

**Figure supplement 1.** Loss of TDP-43 in oligodendrocytes does not result in degeneration of spinal motor neurons.

observed in other cell types and the high incidence of UG repeats recognized by TDP-43 in the genome (*Humphrey et al., 2017*). The differential gene expression profiles and gene ontology analysis were also distinct between the mutant mouse lines (*Figure 6B*; *Figure 6—figure supplement 2*), with early deletion resulting in changes in mRNA processing and cell localization, and latter deletion resulting in changes in regulation of gene expression and metabolic processes.

Analysis of the top 1% of the highest expressed genes enriched in oligodendrocytes, many of which are involved in oligodendrocyte development and myelination, revealed that cryptic exons were often incorporated in their mRNAs (*Figure 6F*). These included *Ermn* (Ermin), an oligodendrocyte-specific cytoskeletal protein that regulates process branching and sheath extension (*Brockschnieder et al., 2006*). TDP-43 loss resulted in the emergence of a novel cryptic exon in intron 2 of the *Ermn* transcript (*Figure 6C*). RT-PCR validation using whole mouse brain indicated that this aberrant splicing generated novel splice isoforms, arising from cryptic exon incorporation, exon extension, and truncation (*Figure 6D*) that may lead to both gain and loss of function. Total Ermin protein was significantly lower in Mobp-TDP43-RCE KO brains compared to controls (*Figure 6E*), and immunohistochemical staining revealed that Ermin was no longer detected in oligodendrocyte somata in these animals (*Figure 6—figure supplement 3*). Thus, TDP-43 deficiency leads to an early missplicing of *Ermn* mRNA, resulting in a partial loss of functional protein. These results indicate that progressive loss of TDP-43 from oligodendrocytes in the adult CNS leads to altered splicing of key regulators of oligodendrocyte growth and morphogenesis, disrupting myelin patterns, inducing gliosis and impairing neuronal function.

## Discussion

New oligodendrocytes are generated in the adult CNS to modify circuits and restore myelin lost through injury or disease (*Gibson et al., 2014*; *Orthmann-Murphy et al., 2020*). This oligodendrogenesis requires transformation of highly dynamic, proliferative progenitors into stable, postmitotic oligodendrocytes that exhibit extraordinary longevity (*Tripathi et al., 2017*). The profound structural and functional changes experienced by oligodendrocyte lineage cells are orchestrated by a family of RNA-binding proteins that ensure transcripts are spliced correctly and delivered to their appropriate location within the cell prior to translation (*Müller et al., 2013*). Cellular stress or direct mutation of these genes can lead to mislocalization and progressive loss of their function, contributing to cellular pathology in diverse neurodegenerative diseases (*Kapeli et al., 2017*; *Purice and Taylor, 2018*), but the consequences of these effects within oligodendroglia are not well understood. As oligodendrocyte lineage cells continue to exist in a dynamic developmental continuum in the adult CNS, these changes may influence progenitor homeostasis, lineage progression, cellular survival, and function, depending on the stage at which gene function is disrupted.

To genetically interrogate the role of TDP-43 within different stages of oligodendrocyte maturation, we employed four different Cre and CreER mouse lines: (1) *Pdgfra-CreER*, (2) *Mobp-iCre*, (3) *Mogi^iCre^*, and (4) *Mobp-iCreER*, allowing *Tardbp* inactivation at discrete stages within the oligodendrocyte lineage in both the developing and mature CNS. Bulk RNA-Seq of purified oligodendrocytes from *Mobp-iCre* and *Mogi^iCre^* mouse lines showed that this approach was able to target different stages of oligodendrocyte development. In particular, oligodendrocytes labeled by *Mobp-iCre* showed higher expression of genes enriched in newly formed oligodendrocytes, such as *Gpr37* and *Mag*, whereas those labeled by *Mogi^iCre^* had higher expression of genes enriched in myelin-forming and mature oligodendrocytes,

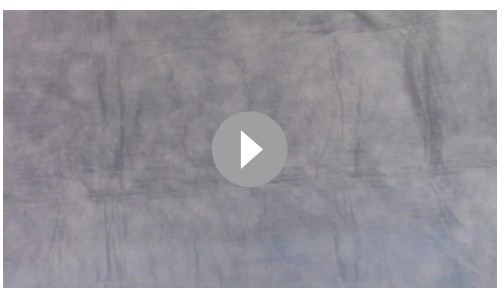

**Video 4.** Video of Mobp-TDP43 Control and cKO mice where Mobp-TDP43 cKO mice exhibit hindlimb paralysis, limp tail, and hindlimb clasping.
https://elifesciences.org/articles/75230/figures#video4

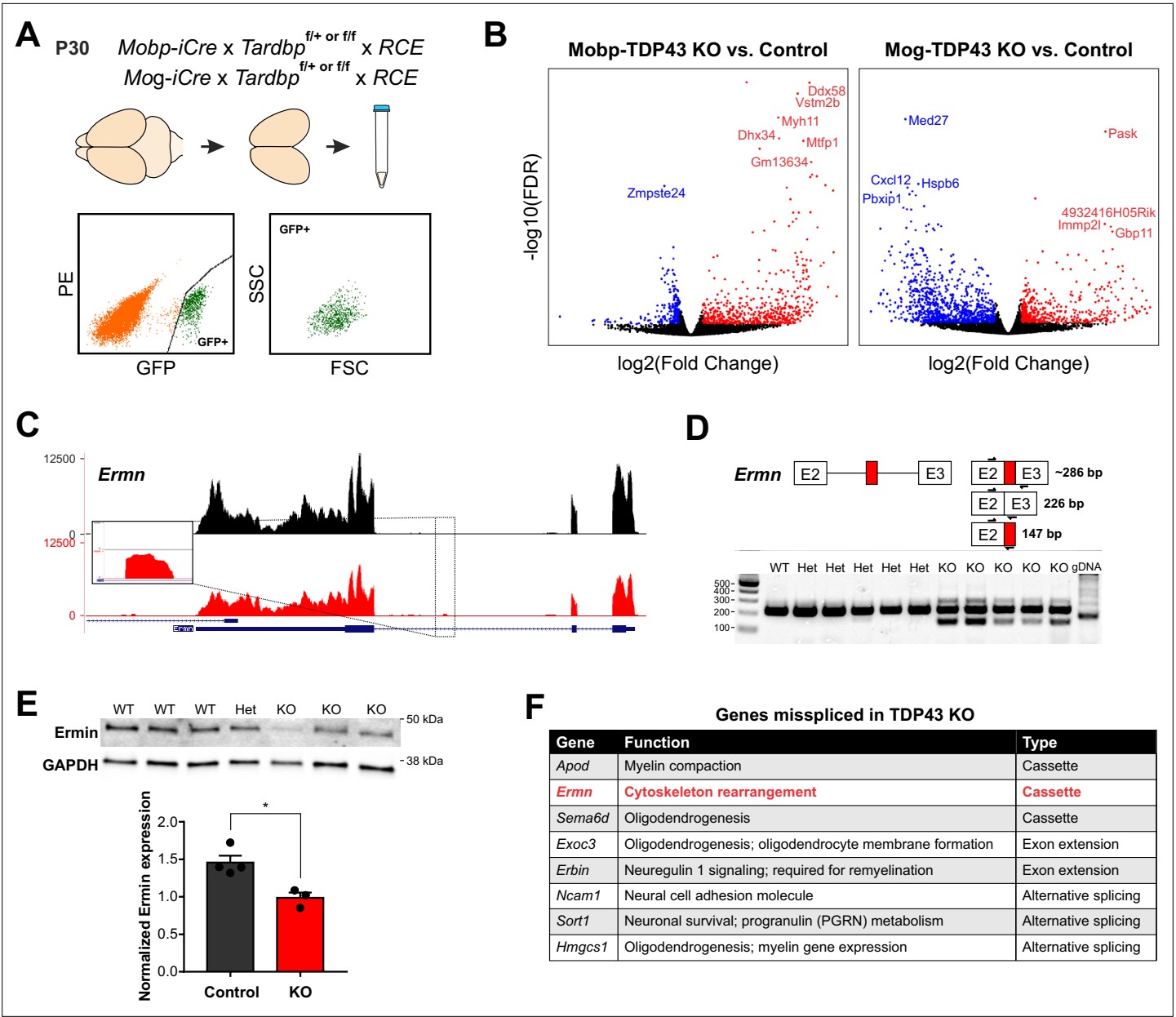

**Figure 6.** TDP-43 maintains oligodendrocyte transcriptional integrity by suppressing cryptic exon incorporation in key oligodendrocyte genes. (**A**) Bulk RNA-Seq was performed with fluorescence-activated cell sorting (FACS)-isolated EGFP+ oligodendrocytes from P30 Mobp-TDP43-RCE and Mog-TDP43-RCE cortices. (**B**) Volcano plots comparing KO to Control in Mobp-TDP43 and Mog-TDP43 mouse lines showing statistically significantly differentially expressed genes (fold change >1, adjusted p value [false discovery rate, FDR] <0.05). Blue dots indicate downregulated genes whereas red dots indicate upregulated genes with TDP-43 KO. (**C**) Visualization of the cryptic exon located in *Ermn*. Gene annotation is shown on the bottom, labeling exons (thick) and introns (thin). Bulk RNA-Seq reads from Control and KO oligodendrocytes from both Mobp-TDP43 and Mog-TDP43 mouse lines are aligned to the mm10 genome. The cryptic exon in the intron 2 of *Ermn* is magnified to highlight the difference. (**D**) Reverse transcription-polymerase chain reaction (RT-PCR) validation of cryptic exon incorporation in *Ermn* in mouse whole brain tissue from Mog-TDP43 Control and KO. Three primers were used to identify (1) normal transcript (226 bp), (2) cryptic exon incorporated transcript (286 bp), and (3) the presence of cryptic exon (147 bp). Only the KO samples showed bands at 147 and 286 bp that represent cryptic exon incorporation into *Ermn* mRNA. (**E**) Western blotting assay to quantify the amount of Ermin protein in Mog-TDP43 Control and KO brains shows that Mog-TDP43 KO brains have a significantly decreased amount of Ermin protein (unpaired, two-tailed Student's *t*-test, *p value = 0.0119, *n* = 4 and 3). (**F**) Table of genes that are misspliced in oligodendrocytes with loss of TDP-43. Cassette indicates incorporation of a nonconserved, novel cryptic exon whereas alternative splicing means usage of alternative, conserved exons. Exon extension indicates that the intronic sequence following an exon was incorporated into mature mRNA and became part of the exonic sequence.

The online version of this article includes the following source data and figure supplement(s) for figure 6:

*Figure 6 continued on next page*

*Figure 6 continued*

**Source data 1.** Raw, cropped, and annotated images of the western blot of mouse-Ermin (red; IRDye 680RD) and rabbit-GAPDH (green; IRDye 800CW).

**Figure supplement 1.** The number of genes that are statistically significantly differentially expressed between different genotypes (false discovery rate FDR <0.05).

**Figure supplement 2.** Gene ontology (GO) analysis from differentially expressed genes shows that Mobp-TDP43 KO and Mog-TDP43 KO show differential enrichment of biological processes.

**Figure supplement 3.** Immunostaining for Ermin shows that KO oligodendrocytes have reduced cell body expression of Ermin protein compared to the Control.

**Figure supplement 4.** Fragments per kilobase of exon per million mapped fragments (FPKM) of *Gpr37*, *Mag*, *Gamt*, and *Opalin* in Mobp-TDP43 Control and Mog-TDP43 Control from bulk RNA-Seq in *Figure 6*.

**Figure supplement 5.** Summary diagram of how TDP-43 exerts a stage-specific role in oligodendrocyte lineage cells.

such as *Gamt* and *Opalin* (*Figure 6—figure supplement 4*). Therefore, these two mouse lines, in combination with *Pdgfra-CreER* mice to manipulate OPCs (*Kang et al., 2010*), allowed specific interrogation of TDP-43 function at distinct developmental stages, revealing a differential requirement for TDP-43 in cell survival and structural maintenance as OPCs transform into myelinating oligodendrocytes (*Figure 6—figure supplement 5*).

## Homeostasis promotes rapid recovery after OPC depletion

Inducible genetic deletion of *Tardbp* within OPCs resulted in their rapid degeneration, with loss first detected in the white matter of the corpus callosum 2 weeks after tamoxifen exposure, followed closely by OPC depletion in cortical gray matter, demonstrating that these progenitors require TDP-43 for survival. OPCs represent the largest population of actively dividing cells in the brain outside the subventricular and subgranular germinal zones (*Dawson et al., 2003*), and maintain a constant density through homeostatic proliferation, in which neighboring cells are induced to divide to replace cells lost through death or differentiation (*Hughes et al., 2013*). Proliferative cells in many tissues are particularly vulnerable to perturbations in RNA processing, as they undergo metabolic reprogramming requiring extensive transcriptional reorganization (*Petasny et al., 2021*). Thus, death of only a few proliferating OPCs upon removal of TDP-43 is likely to induce a cascading cellular depletion, in which neighboring cells (also TDP-43 deficient) are induced to proliferate but then die, inducing further proliferation of remaining cells and continuing until the entire population of TDP-43-deficient OPCs is depleted. The higher basal rate of OPC proliferation in white matter, reflecting their higher rate of differentiation (*Baracskay et al., 2002*; *Viganò et al., 2013*), may be responsible for more rapid loss of OPCs in this region.

Despite decimation of OPCs after TDP-43 deletion, they were restored to their original density in both gray and white matter within 30 days. Because *Tardbp* was not inactivated in all OPCs (the proportion limited by the efficiency of tamoxifen-induced recombination), surviving cells that retain TDP-43 are expected to proliferate to reconstitute the population. Consistent with this hypothesis, all OPCs after 30 days of recovery were EGFP negative (*Figure 1—figure supplement 1D*), indicating that they did not experience Cre activity. These results imply that focal loss of OPCs arising from deficits in TDP-43 should not lead to persistent depletion. It is also possible that the regenerated OPCs were derived from the neural progenitor pool in the subventricular zone, as previously described in the context of remyelination (*Xing et al., 2014*). Indeed, OPCs have been shown to be exceptionally resistant to perturbations in their distribution and density (*Xing et al., 2021*; *Dang et al., 2019*), and rapidly restore their numbers after mass differentiation following demyelination in mice (*Baxi et al., 2017*; *Penderis et al., 2003*). Nevertheless, reductions in OPC density have been observed in some human MS lesions (*Chang et al., 2000*), suggesting that other environmental changes, such as persistent inflammation or gliosis, which may induce TDP-43 mislocalization, may impede their homeostasis and impair their ability to detect demyelinated axons and generate new oligodendrocytes.

## Stage-dependent oligodendrocyte survival in the absence of TDP-43

The differentiation of OPCs into mature, myelinating oligodendrocytes is associated with dramatic structural and functional changes, in which highly motile OPCs stabilize, elaborate processes, and select targets for myelination (*Kirby et al., 2006*; *Almeida et al., 2011*). Previous studies of *Tardbp*

inactivation using *Cnp^Cre* mice (*Wang et al., 2018*; *Chang et al., 2021*), which recombines in the late progenitor and early premyelinating stages, indicate that loss of TDP-43 during this transition period leads to RIPK1-mediated necroptosis of oligodendrocytes (*Wang et al., 2018*). We used two constitutive Cre lines (*Mobp-iCre* and *Mog^iCre*) expressed at different stages (early and late) of the maturation of postmitotic oligodendrocytes to determine whether the effects of TDP-43 loss depend on their state of differentiation. In contrast to TDP-43-deficient OPCs, oligodendrocytes were generated, and myelin sheaths were formed in both lines despite inactivation of *Tardbp*. As the half-life of TDP-43 in vitro was measured to be 12–34 hr (*Ling et al., 2010*; *Lee et al., 2011*), it is possible that sufficient residual TDP-43 RNA/protein remains from the OPC stage to enable these initial transitions. Although oligodendrocytes and myelin were generated, TDP-43-deficient oligodendrocytes formed sheaths with fewer wraps and overall fewer axons were myelinated, similar to that observed after *Cnp^Cre*-mediated deletion (*Wang et al., 2018*). Reconstructions of individual oligodendrocytes at this stage revealed that fewer sheaths were formed per cell on average, indicating that the overall myelin output of these newly formed oligodendrocytes is compromised in the absence of TDP-43. However, the ultimate outcome of these manipulations was distinct, with early deletion (using *Mobp-iCre*) resulting in oligodendrocyte degeneration, seizures, and premature death by 6 months of age. This phenotype is similar to, but not as severe as that observed in *Cnp^Cre*;*Tardbp^{fl/fl}* mice, in which mature oligodendrocytes, OPCs, premyelinating oligodendrocytes, and Schwann cells were genetically manipulated (*Wang et al., 2018*; *Chang et al., 2021*), indicating that disruption of oligodendrocyte TDP-43 levels is sufficient to enhance neuronal excitability and induce premature death.

Although oligodendrocyte density remained stable after *Mobp-iCre*-mediated *Tardbp* inactivation, reactive gliosis occurred throughout the CNS and there was an increase in premyelinating oligodendrocytes (lncOL1+) and enhanced proliferation of OPCs, indicative of oligodendrocyte degeneration, similar to the phenotypic changes observed after inducible deletion of oligodendrocytes in the adult CNS (*Traka et al., 2015*; *Oluich et al., 2012*). Previous studies have shown that subtler manipulations of mature oligodendrocytes, such as reduction in expression of the inwardly rectifying potassium channel $K_{ir}4.1$ (*Larson et al., 2018*) or deletion of the lactate transporter MCT1 (*Philips et al., 2021*), is sufficient to alter neuronal excitability, highlighting possible functional consequences of TDP-43 loss from these myelinating cells.

Unexpectedly, inactivation of *Tardbp* from oligodendrocytes at a later developmental stage (with *Mog^iCre*) led to a strikingly different outcome: oligodendrocytes survived, OPC proliferation was not enhanced and there was no gliosis. Correspondingly, these animals also did not exhibit seizures or premature death. Bulk RNA-Seq transcriptional analysis of oligodendrocytes with early and late deletion, even prior to the phenotypic changes in *Mobp-iCre* mice (P30), revealed that pathways involved in gene expression, cellular localization, cell adhesion, and metabolism were affected. Further, differential gene expression analysis of early, premyelinating and myelinating oligodendrocytes from obtained through single-cell RNA sequencing (*Hrvatin et al., 2018*) revealed that different transcription factors, such as *Mycl*, and transporters, such as MCT1 (*Slc16a1*), change in gene expression as oligodendrocytes mature (*Figure 2—figure supplement 4*). Thus, rapid changes in transcriptional activity during cellular maturation can profoundly alter the impact of TDP-43 depletion, with deficits occurring earlier in maturation leading to more severe outcomes.

The phenotypes observed in response to deletion of TDP-43 from newly generated oligodendrocytes, including subsequent seizures and premature death, may also reflect developmental changes in neural circuits, as this manipulation inactivated *Tardbp* during a critical phase of brain maturation. Indeed, deletion of TDP-43 from oligodendrocytes in the adult CNS (using *Mobp-iCreER*) led to distinct pathological changes, such as hindlimb weakness, hindlimb clasping, and limp tail, phenotypes not observed with developmental *Tardbp* inactivation, and these mice did not exhibit seizures or premature death. Although these distinct phenotypes may arise from differences in the proportion of oligodendrocytes affected, they mirror functional differences observed in neurodevelopmental and neurodegenerative diseases. In neurodevelopmental disorders, such as autism spectrum disorders and schizophrenia, abnormal myelination and altered density of oligodendrocytes have been correlated with neural hyperexcitability and circuit dysfunction (*Phan et al., 2020*; *Fields, 2008*). In contrast, oligodendrocyte turnover and demyelination in the adult CNS have been shown to be accompanied by inflammation and neuronal death in neurodegenerative diseases, such as ALS and AD (*Kang et al., 2013*; *Desai et al., 2010*; *Behrendt et al., 2013*). These results suggest that the consequences of

TDP-43 loss within the oligodendrocyte lineage depends not only on their stage of maturation, but also the environment in which it occurs, with progressive loss in the adult CNS sufficient to impair motor coordination, a hallmark of neurodegenerative diseases such as ALS.

## Aberrant target ensheathment in the absence of TDP-43

In vitro studies indicate that oligodendrocytes have an intrinsic ability to wrap structures of appropriate caliber (>0.4 µm), including inert nanofibers and fixed axons (*Rosenberg et al., 2008*; *Mei et al., 2013*; *Lee et al., 2012*). However, oligodendrocytes in vivo exhibit exquisite target specificity, forming myelin exclusively around axons of appropriate diameter, while avoiding similarly sized dendrites and other nonneuronal structures, such as glial cell processes and capillaries. Moreover, after an initial period of remodeling, the array of myelin sheaths formed by each oligodendrocyte is extremely stable, in terms of position and length along axons. Deletion of TDP-43 from oligodendrocytes either early or late in their maturation resulted in striking, progressive changes in oligodendrocyte morphology, including extension of new processes and wrapping of inappropriate targets, such as neuronal somata and capillaries. Inappropriate myelination of neuronal somata has been observed after deletion of adhesive proteins (e.g., JAM2) from neurons (*Redmond et al., 2016*), expression of a mutant form of a cell adhesion protein (e.g., Cadm4) in oligodendrocytes (*Elazar et al., 2019*), or overexpression of MBP (*Almeida et al., 2018*). Similar aberrant myelination has been recently described in MS lesions (*Neely et al., 2022*), where TDP-43 mislocalization has also be observed (*Masaki et al., 2020*). The morphological changes exhibited by TDP-43-deficient oligodendrocytes suggests that they revert to a progrowth state (*Grinspan et al., 1993*; *Lee et al., 2019*), allowing them to overcome repulsive cues or lack of adhesion that normally prevents wrapping of nonaxonal structures.

## Loss of TDP-43 leads to aberrant RNA splicing

TDP-43 maintains the integrity of the cellular transcriptome by regulating splicing of pre-mRNAs and suppressing incorporation of cryptic exons. By efficiently binding to UG microsatellite repeats within intronic sequences, TDP-43 ensures that spliceosomes skip over these regions during normal splicing (*Ling et al., 2015*). As the specificity of TDP-43 targets primarily depend on the location of these UG tandem repeats, TDP-43-dependent cryptic exons are both cell type and species specific (*Jeong et al., 2017*; *Ling et al., 2015*). By performing RNA sequencing of TDP-43-deficient oligodendrocytes purified from the brain, we uncovered a novel set of cryptic exons in mRNAs that encode proteins involved in oligodendrogenesis and myelination, changes that resulted from novel cassette insertion, exon extension, and alternative splicing. In case of Ermin, a cytoskeletal protein involved in oligodendrocyte morphogenesis (*Brockschnieder et al., 2006*), cryptic exon incorporation led to a pronounced decrease in protein expression; however, the phenotypic changes induced by loss of TDP-43 are not the same as that observed in *Ermn* KO mice (*Wang et al., 2020*; *Ziaei et al., 2020*). Aberrant splicing of Ermin mRNA is predicted to lead to a truncated N-terminal peptide (~amino acid 1–110) that may produce a dominant negative effect. Moreover, loss of TDP-43 was not selective for Ermin, but led to aberrant splicing of numerous myelin and oligodendrocyte genes (*Figure 6F*). *Nfasc*, which is misspliced in TDP-43-deficient Schwann cells (*Chang et al., 2021*), was not alternatively spliced in TDP-43-null oligodendrocytes, further confirming the cell-type-specific actions of TDP-43. The combination of gain and loss of function of the resultant proteins likely contribute to complex phenotype exhibited by these mutant oligodendrocytes.

Oligodendrocyte and myelin abnormalities have been reported in many neurodegenerative diseases known to present with TDP-43 proteinopathy (*Kang et al., 2013*; *Masaki et al., 2020*; *Zhang et al., 2019*). Although cytoplasmic aggregates of TDP-43 are not always present, nuclear clearance of TDP-43 and resulting loss of function is sufficient to cause dramatic alterations in pre-mRNA splicing (*Sun et al., 2017*). Further investigation of RNA splicing changes in oligodendrocytes in different neurodegenerative diseases may reveal unique pathological signatures associated with disease states

that may help predict progression and identify novel therapeutic approaches for reducing the impact of impaired TDP-43 activity.

# Materials and methods

**Key resources table**

| Reagent type (species) or resource | Designation | Source or reference | Identifiers | Additional information |
|---|---|---|---|---|
| Strain, strain background (*Mus musculus*, C57BL6) | *Pdgfra-CreER* | *Kang et al., 2010* | | |
| Strain, strain background (*Mus musculus*, C57BL6) | *Tardbp*<sup>floxed</sup> | *Chiang et al., 2010* | | |
| Strain, strain background (*Mus musculus*, Mixed) | RCE (*Rosa-CAG-LSL-EGFP*) | *Sousa et al., 2009* | JAX MMRRC stock #32,038 | |
| Strain, strain background (*Mus musculus*, C57BL6) | *Mobp-iCre* | This paper | | Generated and maintained by D.E. Bergles lab |
| Strain, strain background (*Mus musculus*, C57BL6) | *Mobp-iCreER* | This paper | | Generated and maintained by D.E. Bergles lab |
| Strain, strain background (*Mus musculus*, C57BL6) | *Mog*<sup>iCre</sup> | *Buch et al., 2005* | | |
| Strain, strain background (*Mus musculus*, C57BL6) | STOCK Tg (*Mobp*-EGFP) IN1Gsat/Mmucd | MMRC | RRID:MMRRC_030483-UCD | *Hughes et al., 2018* |
| Antibody | Anti-Aspartoacylase (ASPA) (rabbit polyclonal) | Gentex | RRID:AB_2036283 | (1:1500) |
| Antibody | Anti-GFP (chicken polyclonal) | Aves lab | RRID:AB_2307313 | (1:1500) |
| Antibody | Anti-MBP (mouse monoclonal) | Sternberger | RRID:AB_2564741 | (1:2000) |
| Antibody | Anti-NG2 (guinea pig serum) | Generated in D.E. Bergles lab against entire NG2 protein *Kang et al., 2013* | | (1:10,000) |
| Antibody | Anti-Ki-67 (rabbit polyclonal) | Abcam | RRID:AB_443209 | (1:1000) |
| Antibody | Anti-GFP (goat polyclonal) | Sicgen | RRID:AB_2333099 | (1:7500) |
| Antibody | Anti-MBP (chicken polyclonal) | Aves lab | RRID:AB_2313550 | (1:500) |
| Antibody | Anti-GFAP (rabbit polyclonal) | Agilent | RRID:AB_10013482 | (1:500) |
| Other | Alexa Fluor 647 Isolectin GS-IB$_4$ | Thermo Fisher | | (1:200) |
| Antibody | Anti-Aquaporin 4 (rabbit polyclonal) | Sigma | RRID:AB_1844967 | (1:1500) |
| Antibody | Anti-NeuN (mouse monoclonal) | Millipore | RRID:AB_2298772 | (1:500) |
| Antibody | Anti-Ermin (mouse monoclonal) | Generated by E. Peles Lab | Gift from E. Peles | (1:1000) |
| Antibody | Anti-Caspr (guinea pig polyclonal) | Generated by M. Bhat Lab | Gift from M. Bhat | (1:1500) |
| Antibody | Anti-βIV spectrin (rabbit polyclonal) | Generated by M. Rasband Lab | Gift from M. Rasband | (1:300) |
| Antibody | Anti-ChAT (goat polyclonal) | Millipore | RRID:AB_90650 | (1:500) |
| Antibody | Anti-CC1 (mouse monoclonal) | Millipore | RRID:AB_2242783 | (1:50) |
| Antibody | Anti-GAPDH (rabbit monoclonal) | Cell Signalling | RRID:AB_10622025 | (1:1000) |
| Antibody | Anti-Rabbit IgG conjugated to Cy3 or Cy5 (donkey polyclonal) | Jackson Immuno | RRID:AB_2313568 RRID:AB_2340625 | (1:2000) |
| Antibody | Anti-Mouse IgG conjugated to Cy3 or Cy5 (donkey polyclonal) | Jackson Immuno | RRID:AB_2340817 RRID:AB_2340820 | (1:2000) |

*Continued on next page*

*Continued*

| Reagent type (species) or resource | Designation | Source or reference | Identifiers | Additional information |
|---|---|---|---|---|
| Antibody | Anti-Guinea pig IgG conjugated to Cy3 or Cy5 (donkey polyclonal) | Jackson Immuno | RRID:AB_2340461 RRID:AB_2340477 | (1:2000) |
| Antibody | Anti-Goat IgG conjugated to Alexa 488 or Cy3 (donkey polyclonal) | Jackson Immuno | RRID:AB_2340430 RRID:AB_2340413 | (1:2000) |
| Antibody | Anti-Chicken IgG conjugated to Alexa 488 or Cy5 (donkey polyclonal) | Jackson Immuno | RRID:AB_2340376 RRID:AB_2340347 | (1:2000) |
| Software, algorithm | ZEN Blue/Black | Zeiss | RRID:SCR_013672 | |
| Software, algorithm | Fiji | http://fiji.sc | RRID:SCR_002285 | |
| Chemical compound, drug | Tamoxifen | Sigma | | |
| Chemical compound, drug | Sunflower seed oil | Sigma | | |

## Generation of *Mobp-iCre* and *Mobp-iCreER* mouse lines

iCreER[T2] sequence was inserted into pBACe3.6_RP23-127H7 to generate *Mobp-iCreER*[T2] BAC for injection in C57BL/6 background. Out of five PCR-positive founders, one of the founders yielded F1 and F2 generations where Cre recombination was independent of tamoxifen injection. Therefore, this line was then designated as a constitutive *Mobp-iCre* mouse line (line #10). Of the other founder lines, the line with highest tamoxifen-dependent recombination efficiency was selected and maintained as a conditional *Mobp-iCreER*T2 mouse line (lines #8, 16) (*Figure 2—figure supplement 1*).

## Animal care and use

Both female and male mice were used for experiments and randomly assigned to experimental groups. Mice were maintained on a 12 hr light/dark cycle, housed in groups no larger than 5, and food and water were provided ad libitum. All animal experiments were performed in strict accordance with protocols approved by the Animal Care and Use Committee at Johns Hopkins University. All experiments were initially performed with a sample size of 3 (biological replicates) for quantification and statistical analyses, and power analysis was then performed to determine the minimum sample size to achieve sufficient statistical power to detect the effect of TDP-43 loss in different transgenic mouse lines at different timepoints. The following transgenic mouse lines were used in this study:

> *Pdgfra-CreER* (*Kang et al., 2010*)
> *Tardbp*[floxed] (*Chiang et al., 2010*)
> RCE (*Rosa-CAG-LSL-EGFP*) (JAX MMRRC stock #32038) (*Sousa et al., 2009*)
> Mobp-iCre
> Mobp-iCreER
> *Mog*[iCre] (*Buch et al., 2005*)
> *Mobp-EGFP* (*Hughes et al., 2018*)

## CreER Induction

To induce *Tardbp* genetic deletion and EGFP expression in the *Pdgfra-CreER;Tardbp*[floxed];*RCE* and *Mobp-iCreER;Tardbp*[floxed];*RCE* mice, P90 or P180–240 mice were injected with tamoxifen (Sigma) dissolved in sunflower seed oil (Sigma) and administered by intraperitoneal injection daily for five consecutive days at a dose of 100 mg/kg body weight. Mice were perfused 14, 30, 60, and 90 days from the last day of tamoxifen injection.

## Cranial windows and in vivo two-photon microscopy

Cranial windows were prepared as previously described (*Orthmann-Murphy et al., 2020*). Briefly, 4- to 8-week-old mice were anesthetized with isoflurane (induction, 5%; maintenance, 1.5–2%, mixed with 0.4–1 l/min O$_2$), and their body temperature was maintained at 37°C with a thermostat-controlled heating plate. The skin over the right hemisphere was removed and the skull cleaned. A 3 × 3 mm region of skull over motor cortex was removed using a high-speed dental drill. A piece of cover glass (VWR, No. 1) was placed in the craniotomy and sealed with VetBond (3 M), then dental cement

(C&B Metabond) and a custom metal plate with a central hole was attached to the skull for head stabilization. In vivo imaging sessions began 3 weeks after cranial window procedure in order for the surgery-induced inflammation to subside. During imaging sessions, mice were anesthetized with isoflurane and immobilized by attaching the head plate to a custom stage. Images were collected using a Zeiss LSM 710 microscope equipped with a GaAsP detector using a mode-locked Ti:sapphire laser (Coherent Ultra) tuned to 920 nm.

## Immunohistochemistry

Mice were deeply anesthetized with sodium pentobarbital (100 mg/kg) and perfused transcardially first with 15–20 ml room temperature PBS (pH 7.4) followed by 10 ml ice-cold 4% paraformaldehyde (PFA in 0.1 M phosphate buffer, pH 7.4). CNS tissues were dissected (e.g., optic nerve, brain, and spinal column) and postfixed in 4% PFA at 4°C for 4–6 or 12–20 hr depending on the antibodies to be used for immunohistochemistry. The tissues samples were transferred to 30% sucrose solution (in PBS, pH 7.4) for cryoprotection for at least two overnights before sectioning. Tissue samples were frozen in TissueTek and sectioned at 35 μm thickness on a cryostat (Thermo Scientific Microm HM 550) at −20°C. Tissue sections were collected into a 24-well plate with PBS with 0.1% $NaN_3$ and kept in 4°C until use. Free-floating sections were preincubated in blocking solution (10% normal donkey serum, 0.3% Triton X-100 in PBS, pH 7.4) for 1 or 2 hr at room temperature, then incubated for 24–48 hr at 4°C or room temperature in primary antibody. Secondary antibody incubation was performed at room temperature for 2–4 hr. Sections were mounted on slides with Aqua Polymount (Polysciences). Images were acquired using a confocal laser-scanning microscope (Zeiss LSM 510 Meta; Zeiss LSM 800; Zeiss LSM 880). The following primary antibodies were used in this study: rabbit anti-ASPA (Genetex; RRID_AB_2036283), chicken anti-GFP (Aves lab; RRID:AB_2307313), goat anti-GFP (Sicgen; RRID:AB_2333099), mouse anti-MBP (Sternberger; RRID:AB_2564741), chicken anti-MBP (Aves lab; RRID:AB_2313550), rabbit anti-GFAP (Agilent; RRID:AB_10013482), Alexa Fluor 647 isolectin GS-IB4 (Thermo Fisher), rabbit anti-Aquaporin 4 (Sigma; RRID:AB_1844967), mouse anti-NeuN (Millipore; RRID:AB_2298772), mouse anti-CC1 (Millipore; RRID:AB_2242783), mouse anti-Ermin (Gift from E. Peles), guinea pig anti-Caspr (Gift from M. Bhat), rabbit anti-BIV-spectrin (Gift from M. Rasband), goat anti-ChAT (Millipore; RRID:AB_90650), guinea pig anti-NG2 (Generated in D.E. Bergles lab), and rabbit anti-Ki-67 (Abcam; RRID:AB_443209). Appropriate donkey secondary antibodies conjugated to Alexa Fluor 488, Cy3 or Alexa Fluor 546, and Cy5 or Alexa Fluor 647 were used at 1:1000–2000.

## Transmission electron microscopy (TEM) and *G* ratio quantification

Samples were perfuse fixed with room temperature PBS and 'Karlsson–Schultz' buffer (2.5% glutaraldehyde, 4% formaldehyde and 0.5% NaCl in phosphate buffer pH 7.3) and postfixed at 4°C for two overnights. Subsequent TEM processing was performed according to a previously described protocol (*Weil et al., 2019*). For *G* ratio analysis 10 random micrographs covering an area of ~180 μm² each were taken at a magnification of ×6000 using a LEO912 transmission electron microscope (Carl Zeiss Microscopy, Oberkochen, Germany) with an on-axis 2k CCD camera (TRS, Moorenweis, Germany). For random selection of axons the images were overlaid with a grid and axons on grid crossings were analyzed with a minimum of 100 axons per animal. For *G* ratio determination the area of the axon and the corresponding fiber including the myelin sheath were measured using Fiji/ImageJ (*Schindelin et al., 2012*) and the diameters derived assuming circular areas. The *G* ratios were calculated as a ratio between axonal diameter and total fiber diameter.

## Image processing and analysis

Image stacks and time series were analyzed using FIJI/ImageJ. For figure presentation, image brightness and contrast levels were adjusted for clarity. Density and *G* ratio quantifications were performed by a blinded observer. Longitudinal image stacks were registered using FIJI plugin 'Correct 3D Drift' (*Parslow et al., 2014*; *Orthmann-Murphy et al., 2020*). Quantification of abnormal wrapping was performed on maximum intensity *z*-projection images. Wrapping of neuronal soma was determined by EGFP+ processes wrapping around NeuN+ neuronal somata, whereas that of blood vessels was determined by the 'tube'-like structures made by EGFP+ oligodendrocyte processes.

## Structural analysis of individual cortical oligodendrocytes

Cortices of *Mog*<sup>iCre</sup>;*Tardbp*<sup>floxed</sup>;*Mobp*-EGFP mice were flatmounted between glass slides, postfixed in 4% PFA for 12 hr at 4°C and transferred to 30% sucrose solution (in PBS, pH 7.4). Tissue was stored at 4°C for 48+ hr before sectioning. Flatmounted hemicortex were extracted, frozen in TissueTek, and sectioned at 40 μm thickness on a cryostat (Thermo Scientific Microm HM 550) at −20°C. Free-floating immunohistochemistry against MBP and EGFP was performed on the first section of the flatmount, which consists of the layer I of the cerebral cortex, as described previously (*Orthmann-Murphy et al., 2020*). High-resolution images were acquired using a confocal laser-scanning microscope (Zeiss LSM 880). All processes originating from the cell body, branch points, and individual myelin sheaths stained with EGFP and MBP were traced in FIJI using Simple Neurite Tracer (*Longair et al., 2011*). A smoothing function was used on the traced segments prior to length calculation to reduce tracing artifacts.

## Clinical EAE behavioral scoring

Clinical EAE behavioral scores were obtained in a masked manner based on the preacquired videos of the animals, using the established standard scoring from 1 to 5; 0 = no signs of disease; 1 = loss of tail tonicity; 2 = loss of tail tonicity and mild paralysis of hindlimbs; 3 = paralysis of hindlimbs; 4 = hindlimbs paralysis and mild paralysis of forelimbs; and 5 = complete paralysis or death (*Gharagozloo et al., 2021*).

## Statistical analysis

Statistical analyses were performed with GraphPad Prism 7 or Excel (Microsoft). All technical replicates (e.g., different brain sections for IHC) for each sample were averaged as a single data point. Significance was determined using one-way ANOVA test with Tukey's correction for multiple comparisons or unpaired, two-tailed Student's *t*-test. Each figure legend otherwise contains the statistical tests used to measure significance and the corresponding significance level (p value). n represents the number of animals (biological replicates) used in each experiment. Data are reported as mean ± standard error of the mean and p value <0.05 was considered statistically significant.

## Fluorescence-activated cell sorting

Male and female Mobp-TDP43 and Mog-TDP43 mice aged at P30 were cardiac perfused with ice-cold 1× Hank's Balanced Salt Solution (HBSS) w/o $Ca^{2+}$ and $Mg^{2+}$. Only the cortex was dissected out for downstream processing and mechanically dissociated with a razor blade on ice. The Miltenyi Neural Tissue Dissociation kit was used to perform enzymatic dissociation of the tissue into a single-cell suspension, and Myelin Debris Removal solution (Miltenyi) to remove myelin debris. The cells were resuspended in FACS sorting buffer (5 ml of 1 M HEPES (4-(2-hydroxyethyl)-1-piperazineethanesulfonic acid) + 2 ml of Fetal Bovine Serum (FBS) + 800 μl of 0.5 M EDTA (Ethylenediaminetetraacetic acid) in 200 ml HBSS w/o $Ca^{2+}$, $Mg^{2+}$) and filtered through 35 μm strainer. The cells were sorted for endogenous GFP fluorescence on BD FACS Aria Ilu Cell Sorter using a 100 μm nozzle at the Ross Flow Cytometry Core Facility at Johns Hopkins University School of Medicine.

## RNA isolation, cDNA library preparation, and sequencing

RNA was isolated from FACS purified oligodendrocytes (~5000 cells/sample) using QIAGEN RNeasy Micro Kit following the manufacturer's instructions. cDNA was synthesized by using QIAGEN QuantiTect Reverse Transcription Kit. Due to low amount of cDNA, Nugen Single-cell RNAseq kit was used to generate cDNA libraries for 100 bp paired-end sequencing on Illumina HiSeq (70–100 million paired reads/sample).

## Analysis of bulk RNA-Seq data

Reads were aligned to the latest mouse mm10 reference genome using the STAR spliced read aligner with default parameters. Total counts of read fragments aligned to known gene regions within the mouse mm10 RefSeq reference annotation were used as the basis for quantification of gene expression. Fragment counts were derived using HTS-seq program using mm10 Ensembl transcripts as the model. Various QC analyses were conducted to assess the quality of the data and to identify potential outliers (e.g., the level of mismatch rate, mapping rate to the whole genome, repeats, chromosomes,

and key transcriptomic regions). Principal component analysis correction was performed to correct for batch effects and differences in sample RNA integrity number. Differentially expressed genes were identified using EdgeR Bioconductor package and ranked based on adjusted p values (false discovery rate) of <0.05.

## RT-PCR

RNA was isolated from whole mouse brain tissue using ReliaPrep RNA Tissue Miniprep System (Promega) following the manufacturer's instructions. cDNA was synthesized by using QIAGEN QuantiTect Reverse Transcription Kit. RT-PCR primers against normal *Ermn* and cryptic exon-specific *Ermn* transcripts were designed using Benchling and Primer3.

## Western blotting

Whole mouse brain tissue was homogenized in 1.5 ml Eppendorf tube with a handheld homogenizer in Pierce RIPA buffer (Thermo Fisher) with Sodium Orthovanadate (Vanadate). After centrifugation at 4°C, supernatant was collected, and the protein concentration was determined using Pierce BCA Protein Assay Kit (Thermo Fisher). 50 µg of protein was loaded into each lane of Mini-Protean TGX gels (BioRad) and ran at 120 V for 1.5 hr. Protein was transferred onto Immobilon-FL PVDF Membrane (Millipore) at 100 V for 1.5 hr. Intercept (TBS) Blocking Buffer (Li-Cor) was used for blocking, and the membrane was incubated in primary antibody solution at 4°C overnight with gentle shaking. Mouse-Ermin (Mab160; raised against Ermin residues 1–261) and Rabbit-GAPDH (Cell Signaling) primary antibodies were used. IRDye 680RD Goat anti-Mouse IgG (H + L) (Li-Cor) and IRDye 800CW Goat anti-Rabbit IgG (H + L) (Li-Cor) were used as secondary antibodies for imaging on Odyssey Imaging Workstation (Li-Cor).

## Acknowledgements

We thank M Pucak at the Multiphoton Imaging (MPI) Core in the Department of Neuroscience at Johns Hopkins for technical assistance, E Peles for gift of the mouse anti-Ermin primary antibody, M Rasband for anti-BIV-spectrin primary antibody, M Bhat for guinea pig anti-Caspr primary, and members of the Bergles lab for discussions. V Doze assisted in quantifying the myelinated axons in TEM images, and CL Call generated a diagram for *Figure 3—figure supplement 3B*. D Heo was supported by NINDS National Research Service Award (NRSA) predoctoral training fellowship (F31NS110204). GC Molina-Castro was supported by a National Science Foundation Graduate Research Fellowship (DGE-1746891). K-A Nave was supported by an ERC Advanced Grant (MyeliNANO) and DFG-TRR274. Funding was provided by grants from The National MS Society, Target ALS and the Dr. Miriam and Sheldon G Adelson Medical Research Foundation to DE Bergles and K-A Nave.

## Additional information

### Competing interests

Wiebke Möbius: Reviewing editor, *eLife*. The other authors declare that no competing interests exist.

### Funding

| Funder | Grant reference number | Author |
|---|---|---|
| National Institutes of Health | R01 AG072305 | Dwight E Bergles |
| National Multiple Sclerosis Society | | Dwight E Bergles |
| National Institutes of Health | F31NS110204 | Dongeun Heo |
| European Research Council | MyeliNANO | Klaus-Armin Nave |

| Funder | Grant reference number | Author |
|---|---|---|
| Deutsche Forschungsgemeinschaft | DFG-TRR274 | Klaus-Armin Nave |
| Target ALS | | Dwight E Bergles |
| Dr. Miriam and Sheldon G Adelson Medical Research Foundation | | Dwight E Bergles |
| Max-Planck-Institute of Experimental Medicine | open access funding | Wiebke Möbius |

The funders had no role in study design, data collection, and interpretation, or the decision to submit the work for publication.

## Author contributions

Dongeun Heo, Data curation, Formal analysis, Methodology, Validation, Visualization, Writing – review and editing, Investigation, Project administration, Resources, Supervision, Writing - original draft; Jonathan P Ling, Formal analysis, Methodology, Writing – review and editing, Project administration, Resources, Writing - original draft; Gian C Molina-Castro, Wiebke Möbius, Formal analysis, Methodology, Visualization, Writing – review and editing, Resources, Writing - original draft; Abraham J Langseth, Methodology, Writing – review and editing, Funding acquisition; Ari Waisman, Funding acquisition, Writing - original draft; Klaus-Armin Nave, Formal analysis, Writing – review and editing, Conceptualization, Investigation, Writing - original draft; Phil C Wong, Data curation, Writing – review and editing, Investigation, Writing - original draft; Dwight E Bergles, Data curation, Validation, Visualization, Writing – review and editing, Conceptualization, Funding acquisition, Investigation, Resources, Supervision, Writing - original draft

## Author ORCIDs

Dongeun Heo ![ORCID] http://orcid.org/0000-0002-4913-2253
Jonathan P Ling ![ORCID] http://orcid.org/0000-0003-1927-9729
Gian C Molina-Castro ![ORCID] http://orcid.org/0000-0002-0700-4042
Klaus-Armin Nave ![ORCID] http://orcid.org/0000-0001-8724-9666
Wiebke Möbius ![ORCID] http://orcid.org/0000-0002-2902-7165
Dwight E Bergles ![ORCID] http://orcid.org/0000-0002-7133-7378

## Ethics

This study was performed in strict accordance with the recommendations by the Institutional Animal Care and Use Committee (IACUC) of the Johns Hopkins School of Medicine under protocols (MO17M338, MO17M268, MO20M206, and MO20M344). All survival surgery was performed under isoflurane anesthesia, and every effort was made to minimize suffering. All terminal experiments were carried out under sodium pentobarbital anesthesia.

## Decision letter and Author response

Decision letter https://doi.org/10.7554/eLife.75230.sa1
Author response https://doi.org/10.7554/eLife.75230.sa2

## Additional files

### Supplementary files

• Supplementary file 1. Differential gene expression of TDP-43 KO oligodendrocytes. Excel file of bulk RNA-Seq that includes average FPKM of each gene, log fold change (FC), p value, and false discovery rate (FDR) for the following comparisons: (1) Mog-TDP43-WT vs. Mobp-TDP43-WT, (2) Mog-TDP43-KO vs. Mobp-TDP43-KO, (3). Mog-TDP43-KO vs. Mog-TDP43-WT, (4) Mobp-TDP43-KO vs. Mobp-TDP43-WT, and (5) TDP43-WT vs. TDP43-KO.

• Transparent reporting form

## Data availability

Bulk RNA-seq data of P30 FACS-isolated oligodendrocytes from Mobp-TDP43 and Mog-TDP43 mouse lines has been deposited to GEO (GSE188903). Processed data, including the raw count number, normalized counts, and FPKM values, are provided as Supplementary Data (Supplementary file 1. Differential gene expression of TDP-43 KO oligodendrocytes).

The following dataset was generated:

| Author(s) | Year | Dataset title | Dataset URL | Database and Identifier |
|---|---|---|---|---|
| Heo D, Ling JP, Molina-Castro GC, Langseth AJ, Waisman A, Nave K-A, Möbius W, Wong PC, Bergles DE | 2021 | Stage-specific control of oligodendrocyte survival and morphogenesis by TDP-43 | https://www.ncbi.nlm.nih.gov/geo/query/acc.cgi?acc=GSE188903 | NCBI Gene Expression Omnibus, GSE188903 |

The following previously published datasets were used:

| Author(s) | Year | Dataset title | Dataset URL | Database and Identifier |
|---|---|---|---|---|
| Hrvatin S, Hochbaum DR, Nagy MA, Cicconet M, Robertson K, Cheadle L, Zilionis R, Ratner A, Borges-Monroy R, Klein AM, Sabatini BL, Greenberg ME | 2018 | Single-cell analysis of experience-dependent transcriptomic states in the mouse visual cortex | https://doi.org/10.1038/s41593-017-0029-5 | NCBI-GSE102827, 10.1038/s41593-017-0029-5 |
| Marques S, Zeisel A, Codeluppi S, van Bruggen D, Falcao AM, Xiao L, Li H, Haring M, Hochgerner H, Romanov RA, Gyllborg D, Manchado AM, Manno GL, Lonnerberg P, Floriddia EM, Rezayee F, Ernfors P, Arenas E, Hjerling-Leffler J, Harkany T, Richardson WD, Linnarsson S, Castelo-Branco G | 2016 | Oligodendrocyte heterogeneity in the mouse juvenile and adult central nervous system | https://www.ncbi.nlm.nih.gov/geo/query/acc.cgi?acc=GSE75330 | NCBI Gene Expression Omnibus, GSE75330 |

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
