## [Editor Report]

Heo et al., 2021 investigated the role of the DNA/RNA-binding protein TDP-43 on oligodendrocyte maturation and showed that deletion of this protein induces deleterious effects on oligodendrocyte function in transgenic mouse models. These findings are exciting and support TDP-43 as a key regulator of oligodendrocyte function which may go awry in neurodegenerative disorders.

---

## [Decision Letter]

**Decision letter after peer review:**

Thank you for submitting your article "Stage-specific control of oligodendrocyte survival and morphogenesis by TDP-43" for consideration by *eLife*. Your article has been reviewed by 3 peer reviewers, one of whom is a member of our Board of Reviewing Editors, and the evaluation has been overseen by Gary Westbrook as the Senior Editor. The following individual involved in review of your submission has agreed to reveal their identity: Maria D Purice (Reviewer #3).

The reviewers have discussed their reviews with one another, and the Reviewing Editor has drafted this to help you prepare a revised submission with specific attention to the essential revisions.

Essential revisions:

The majority of the comments are focused on the clarity of the written document, as well as suggestions for making the figures more informative as well as specific recommendations for increasing the clarity of the figures and providing quantification of a few of the figures.

– There were a few additional experimental data requested centered on reporter data on recombination efficiency (see reviewer 2 point #3)- and neurofascin splicing (see rev 2 point 5).

– Please provide more details and quantification of the immunohistochemistry images shown as noted by reviewer 1 in legends and methods (in particular Figures 4, 5).

– Please address reviewer 3 comment regarding potential loss of OPC identity. All reviewers agreed this is important especially since one marker (NG2) was used to define OPCs. Because the absence of TDP-43 alters splicing, it is possible that NG2 is not being expressed. This could also help explain the lack of astrogliosis.

*Reviewer #1 (Recommendations for the authors):*

Major Strengths:

• Mog-TDP43 and Mobp-TDP43 KO mice represent useful tools for the field to study early and late stages of oligodendrocyte maturation

• The stage-dependent role of TDP-43 in oligodendrocyte maturation, morphology, and function is particularly novel.

• Transcriptomic analysis (Figure 6) is particularly strong, showing that TDP-43 KO oligodendrocytes exhibit missplicing of cytoskeletal-associated genes

Major Weaknesses:

• Very minimal quantification of immunohistochemistry images (Figure 4 in particular). Quantification of all images must be included with details in methods/legends

• Several figures could be split up into separate figures to enhance clarity. Schematics should also be included in separate figures to provide additional visuals of experiments

• No stain of TDP-43 in immunohistochemistry images. Please address.

• No quantification of behavior/hindlimb weakness (Figure 5)

• Narrative is lost through the vast amount of data shown, especially within each figure

• Figure 3 axon thinning and density could be shown via A,B,D,E,F and without C and G

• Figure 4 does not need examples of abnormal structures (could be put in supplement), and lacks quantification

• Both Figures 3 and 4 could be condensed or moved to supplemental

*Reviewer #2 (Recommendations for the authors):*

The following points are suggestions to improve the overall presentation and impact of the manuscript.

1. In the first section of the results the authors mention assessing the efficiency of Cre-mediated recombination in the PDGFRa-TDP43 mice by measuring EGFP expression in NG2-positive OPCs. It is likely that a sentence got deleted that described PDGFRa-CreER mice mated to EGFP Cre reporter (RCE) mice to allow for this assessment. It is mentioned in the figure 1 legend.

2. The data presented clearly shows that TDP-43 is required for the survival of OPCs in young and older adult animals. Strikingly, the authors also show that the population of OPCs is quickly restored following their loss. The authors make the reasonable suggestion that the repopulated cells are derived from the minor population of OPCs that did not originally delete TDP-43. Nevertheless, it also seems possible that these cells are derived from a "pre-OPC" progenitor pool that through cell division and maturation are able to replenish the depleted OPC pool. The authors' interpretation is based on their extensive, groundbreaking work on how OPCs replenish lost oligodendrocytes, but it might be possible that distinct mechanisms respond to OPC loss. It might be worth mentioning this alternative possibility in the discussion.

3. The authors have done an excellent job demonstrating recombination efficiency and specificity using the PDGFRa and Mobp Cre drivers. Nevertheless, they have not shown similar data for the Mog Cre driver. This data would be helpful when considering the apparent lack of impact on oligodendrocyte viability, in contrast with the other drivers, in the Mog-Cre mice.

4. The sentence starting the new section on page 8 begins "Despite the eventual vulnerability of oligodendrocytes to loss of TDP-43, in both Mobp-TDP43 KO and Mog-197 TDP43 KO mice", which is confusing. The authors have clearly shown oligodendrocyte "vulnerability" in the Mobp animals, but up to this point in the manuscript all of the assays presented appear to show unperturbed oligodendrocytes in the Mog animals. Subsequently they demonstrate thinner myelin in these mice, but the introductory statement should be clarified or re-written.

5. Chang et al. (2021) recently demonstrated that the loss of TDP-43 in myelinating Schwann cells resulted in aberrant splicing of the paranodal component Neurofascin, which is also expressed by oligodendrocytes. The authors of the current study also demonstrate aberrant splicing in the TDP-43 deficient oligodendrocytes. It would be interesting to know whether Neurofascin RNA is also abnormally processed in the oligodendrocyte mutants. Paranodal abnormalities might contribute to the clinical phenotype of the mutant animals.

*Reviewer #3 (Recommendations for the authors):*

Using their mouse models, the authors are able to show the role of TDP-43 in OPC survival, and the role of TDP-43 in early and late stages of postmitotic oligodendrocytes. It would be useful to have a summary figure that shows the role of TDP-43 in OPC/oligodendrocyte development.

In lines 164-66, the authors mention that their results are "similar to the phenotype exhibited by Cnp-Cre;Tardbp 165 fl/fl mice (J. Wang et al. 2018), in which recombination occurs in OPCs, early postmitotic oligodendrocytes, and Schwann cells in the PNS (Lappe-Siefke et al. 2003)." This statement and citations are confusing. In J. Wang et al., 2018 they confirmed that Cre-mediated recombination occurs for over 95% of mature oligodendrocytes, which are labeled with APC-CC1 and they saw no colocalization of GFP with NG2-positive OPCs, thus suggesting that Cnp-Cre-mediated recombination in the spinal cord is restricted to mature oligodendrocytes. In addition, Lappe-Siefke et al. 2003 does not look at Schwann cells in the PNS. Please clarify these citations.

Lines 343-4. The sentence "Thus, death of only a few proliferating OPCs upon removal of TDP-43 is likely to induce a cascading depletion, capturing neighboring cells (also TDP-43 deficient) as they are induced to proliferate" is confusing. Can the authors clarify what they mean by a cascading-depletion of TDP-43 in other neighboring cells?

The methods are easy to follow, however for the RNA isolation section (line 569), it would be helpful to include the number of oligodendrocytes that were sorted and used for RNA extraction.

---

## [Author Response]

Essential revisions:The majority of the comments are focused on the clarity of the written document, as well as suggestions for making the figures more informative as well as specific recommendations for increasing the clarity of the figures and providing quantification of a few of the figures.– There were a few additional experimental data requested centered on reporter data on recombination efficiency (see reviewer 2 point #3)- and neurofascin splicing ( see rev 2 point 5).

The efficiency of Cre recombination in *Mog*-iCre mice was quantified by crossing these mice to a Cre-dependent EGFP reporter mouse line (“RCE”). Analysis of these mice [add experimental details – timing of tam and sac], revealed that 97.3±0.7% of ASPA^+^ oligodendrocytes were EGFP^+^ in the cerebral cortex ( = recombination efficiency), whereas 95.9±0.5% of EGFP^+^ cells were ASPA^+^ ( = recombination specificity). A small percentage of neurons exhibit Cre activity in this animals (P90: 3.2±0.3%, P180: 4.9±0.7%). These data have been added as new Figure 2—figure supplement 1.

We did not observe any changes in *Nfasc* splicing in cortical oligodendrocytes with *Tardbp* deletion. These data are consistent with observations recently reported by Chang *et al.* (2021), in which *Nfasc* missplicing was observed in sciatic nerve, but not in the spinal cord after *Cnp*-Cre-mediated deletion of *Tardbp* (see Figure 6C of Chang et al. 2021). These observations have been added to the Discussion section of the manuscript (p. 18).

– Please provide more details and quantification of the immunohistochemistry images shown as noted by reviewer 1 in legends and methods (in particular Figures 4, 5).

We have added quantification of the prevalence (# of events/µm^3^) of abnormal association of TDP-43 KO oligodendrocytes with non-axonal structures in Figure 4—figure supplement 4. The method of quantification is now described in the Figure 4—figure supplement 4 legend and the Materials and Methods section.

– Please address reviewer 3 comment regarding potential loss of OPC identity. All reviewers agreed this is important especially since one marker (NG2) was used to define OPCs. Because the absence of TDP-43 alters splicing, it is possible that NG2 is not being expressed. This could also help explain the lack of astrogliosis.

We agree this is an important issue. In these experiments, a Cre-dependent EGFP reporter transgene (RCE) was included to follow the fate of Cre-recombined OPCs over time. Immunolabeling for EGFP in these mice revealed that all EGFP^+^ TDP-43 KO OPCs disappear, rather than downregulate NG2. We also show that PDGFRα+ cells also disappear upon Cre recombination. These data have been added to Figure 1—figure supplement 1D. Of note, a previous study that used diphtheria toxin to ablate OPCs reported that widespread apoptosis of OPCs does not induce microglial activation or pro-inflammatory changes (Birey *et al.*, 2015).

Reviewer #1 (Recommendations for the authors):Major Strengths:• Mog-TDP43 and Mobp-TDP43 KO mice represent useful tools for the field to study early and late stages of oligodendrocyte maturation• The stage-dependent role of TDP-43 in oligodendrocyte maturation, morphology, and function is particularly novel.• Transcriptomic analysis (Figure 6) is particularly strong, showing that TDP-43 KO oligodendrocytes exhibit missplicing of cytoskeletal-associated genesMajor Weaknesses:• Very minimal quantification of immunohistochemistry images (Figure 4 in particular). Quantification of all images must be included with details in methods/legends

Quantification of abnormal association with non-axonal structures has been quantified (Figure 4—figure supplement 4). The methods of quantification have been added to the Material and Methods section.

• Several figures could be split up into separate figures to enhance clarity. Schematics should also be included in separate figures to provide additional visuals of experiments

Figure 3 and 4 have been split and a summary figure has been generated (Figure 6—figure supplement 5) outlining the distinct role of TDP-43 at different stages of oligodendrocyte development.

• No stain of TDP-43 in immunohistochemistry images. Please address.

In the mouse brain, TDP-43 protein is most highly expressed by neurons. With available antibodies, immunostaining for TDP-43 only resulted in labeling of neuronal somata. Although TDP-43 is below the limit of detection within oligodendrocyte lineage cells in tissue sections, mRNA for TDP-43 is consistently detected using RNA sequencing approaches and the profound phenotypes we observe indicate its importance in promoting the survival of OPCs, oligodendrocyte maturation, and oligodendrocyte stability.

• No quantification of behavior/hindlimb weakness (Figure 5)

Given the similarity of the motor abnormalities (limp tail, hindlimb weakness) exhibited by TDP43 cKO mice to that observed in mice subjected to experimental autoimmune encephalomyelitis (EAE), we used the well-established EAE 0-5 severity scoring scale to quantify the behavior of these mice. These data are now included in Figure 5E.

• Narrative is lost through the vast amount of data shown, especially within each figure• Figure 3 axon thinning and density could be shown via A,B,D,E,F and without C and G• Figure 4 does not need examples of abnormal structures (could be put in supplement), and lacks quantification• Both Figures 3 and 4 could be condensed or moved to supplemental

We have now separated the main figures (Figures 3 and 4) and moved the data indicated into Supplementary Data.

Reviewer #2 (Recommendations for the authors):The following points are suggestions to improve the overall presentation and impact of the manuscript.1. In the first section of the results the authors mention assessing the efficiency of Cre-mediated recombination in the PDGFRa-TDP43 mice by measuring EGFP expression in NG2-positive OPCs. It is likely that a sentence got deleted that described PDGFRa-CreER mice mated to EGFP Cre reporter (RCE) mice to allow for this assessment. It is mentioned in the figure 1 legend.

Indeed, all of the mice used in this study expressed a Cre-dependent EGFP reporter (RCE) to assess Cre recombination efficiency and to allow visualization of the morphology of TDP-43 KO cells. This information is now included in the Results (p.5).

2. The data presented clearly shows that TDP-43 is required for the survival of OPCs in young and older adult animals. Strikingly, the authors also show that the population of OPCs is quickly restored following their loss. The authors make the reasonable suggestion that the repopulated cells are derived from the minor population of OPCs that did not originally delete TDP-43. Nevertheless, it also seems possible that these cells are derived from a "pre-OPC" progenitor pool that through cell division and maturation are able to replenish the depleted OPC pool. The authors' interpretation is based on their extensive, groundbreaking work on how OPCs replenish lost oligodendrocytes, but it might be possible that distinct mechanisms respond to OPC loss. It might be worth mentioning this alternative possibility in the discussion.

Thank you for this suggestion. Although we favor the conclusion that repopulation occurs primarily from surviving cells, based on the recombination efficiency and qualitative assessments of proliferation of glial progenitors around the lateral ventricles, we now note this possibility in the Discussion (p. 14).

3. The authors have done an excellent job demonstrating recombination efficiency and specificity using the PDGFRa and Mobp Cre drivers. Nevertheless, they have not shown similar data for the Mog Cre driver. This data would be helpful when considering the apparent lack of impact on oligodendrocyte viability, in contrast with the other drivers, in the Mog-Cre mice.

We now provide these data in Figure 2—figure supplement 1.

4. The sentence starting the new section on page 8 begins "Despite the eventual vulnerability of oligodendrocytes to loss of TDP-43, in both Mobp-TDP43 KO and Mog-197 TDP43 KO mice", which is confusing. The authors have clearly shown oligodendrocyte "vulnerability" in the Mobp animals, but up to this point in the manuscript all of the assays presented appear to show unperturbed oligodendrocytes in the Mog animals. Subsequently they demonstrate thinner myelin in these mice, but the introductory statement should be clarified or re-written.

We have removed the text referring to the vulnerability and modified the introductory statement to the following: In both Mobp-TDP43 KO and Mog-TDP43 KO mice, oligodendrocytes were able to mature and form myelin sheaths, global myelin tracts within the brain were established (Figure 3—figure supplement 1A), and there was no evidence of astrogliosis early in development (P30) (Figure 3—figure supplement 2).

5. Chang et al. (2021) recently demonstrated that the loss of TDP-43 in myelinating Schwann cells resulted in aberrant splicing of the paranodal component Neurofascin, which is also expressed by oligodendrocytes. The authors of the current study also demonstrate aberrant splicing in the TDP-43 deficient oligodendrocytes. It would be interesting to know whether Neurofascin RNA is also abnormally processed in the oligodendrocyte mutants. Paranodal abnormalities might contribute to the clinical phenotype of the mutant animals.

We examined our RNAseq datasets and did not observe that *Nfasc* was aberrantly spiced after deletion of *Tardbp*. Of note, Chang *et al.* (2021) observed that aberrant splicing of *Nfasc* occurred in the sciatic nerve but not in the spinal cord in *Cnp*-Cre cKO mice, which recombine in both the CNS and PNS. This result is now reported in the Discussion (p. 18).

Reviewer #3 (Recommendations for the authors):Using their mouse models, the authors are able to show the role of TDP-43 in OPC survival, and the role of TDP-43 in early and late stages of postmitotic oligodendrocytes. It would be useful to have a summary figure that shows the role of TDP-43 in OPC/oligodendrocyte development.

We have added a new summary figure to accompany the manuscript (Figure 6—figure supplement 5).

In lines 164-66, the authors mention that their results are "similar to the phenotype exhibited by Cnp-Cre;Tardbp 165 fl/fl mice (J. Wang et al. 2018), in which recombination occurs in OPCs, early postmitotic oligodendrocytes, and Schwann cells in the PNS (Lappe-Siefke et al. 2003)." This statement and citations are confusing. In J. Wang et al., 2018 they confirmed that Cre-mediated recombination occurs for over 95% of mature oligodendrocytes, which are labeled with APC-CC1 and they saw no colocalization of GFP with NG2-positive OPCs, thus suggesting that Cnp-Cre-mediated recombination in the spinal cord is restricted to mature oligodendrocytes. In addition, Lappe-Siefke et al. 2003 does not look at Schwann cells in the PNS. Please clarify these citations.

We have revised the reference to the following prior studies: Chang et al. 2021 and Tognatta et al. 2017 (p. 7).

Lines 343-4. The sentence "Thus, death of only a few proliferating OPCs upon removal of TDP-43 is likely to induce a cascading depletion, capturing neighboring cells (also TDP-43 deficient) as they are induced to proliferate" is confusing. Can the authors clarify what they mean by a cascading-depletion of TDP-43 in other neighboring cells?

We have clarified this statement (Discussion, p. 14).

The methods are easy to follow, however for the RNA isolation section (line 569), it would be helpful to include the number of oligodendrocytes that were sorted and used for RNA extraction.

This method has been clarified.

References

Ayala YM, De Conti L, Avendano-Vazquez SE, Dhir A, Romano M, D'Ambrogio A, Tollervey J, Ule J, Baralle M, Buratti E, Baralle FE (2011) TDP-43 regulates its mRNA levels through a negative feedback loop. *Embo J* 30:277–288.

Birey, F., Kloc, M., Chavali, M., Hussein, I., Wilson, M., Christoffel, D. J., Chen, T., Frohman, M. A., Robinson, J. K., Russo, S. J., Maffei, A., and Aguirre, A. (2015). Genetic and Stress-Induced Loss of NG2 Glia Triggers Emergence of Depressive-like Behaviors through Reduced Secretion of FGF2. *Neuron*, 88(5), 941–956. https://doi.org/10.1016/j.neuron.2015.10.046.

Chang, Kae Jiun, Ira Agrawal, Anna Vainshtein, Wan Yun Ho, Wendy Xin, Greg Tucker-Kellogg, Keiichiro Susuki, Elior Peles, Shuo Chien Ling, and Jonah R. Chan. 2021. “TDP-43 Maximizes Nerve Conduction Velocity by Repressing a Cryptic Exon for Paranodal Junction Assembly in Schwann Cells.” e*Life* 10 (March). https://doi.org/10.7554/elife.64456.

Hughes, Ethan G., Shin H. Kang, Masahiro Fukaya, and Dwight E. Bergles. 2013. “Oligodendrocyte Progenitors Balance Growth with Self-Repulsion to Achieve Homeostasis in the Adult Brain.” *Nature Neuroscience 16* (6): 668–76. https://doi.org/10.1038/nn.3390.

Igaz LM, Kwong LK, Lee EB, Chen-Plotkin A, Swanson E, Unger T, Malunda J, Xu Y, Winton MJ, Trojanowski JQ, Lee VM (2011) Dysregulation of the ALS-associated gene TDP-43 leads to neuronal death and degeneration in mice. *J Clin Invest* 121:726–738.

Orthmann-Murphy, Jennifer, Cody L Call, Gian Carlo Molina-Castro, Yu Chen Hsieh, Matthew N Rasband, Peter A Calabresi, and Dwight E Bergles. 2020. “Remyelination Alters the Pattern of Myelin in the Cerebral Cortex.” e*Life* 9: 1–61. https://doi.org/10.7554/*eLife*.56621.

Polymenidou M, Lagier-Tourenne C, Hutt KR, Huelga SC, Moran J, Liang TY, Ling SC, Sun E, Wancewicz E, Mazur C, Kordasiewicz H, Sedaghat Y, Donohue JP, Shiue L, Bennett CF, Yeo GW, Cleveland DW (2011) Long pre-mRNA depletion and RNA missplicing contribute to neuronal vulnerability from loss of TDP-43. *Nat Neurosci* 14:459–468.

Tognatta, R., Sun, W., Goebbels, S., Nave, K. A., Nishiyama, A., Schoch, S., Dimou, L., and Dietrich, D. 2017. "Transient Cnp expression by early progenitors causes Cre-Lox-based reporter lines to map profoundly different fates". *Glia* 65(2), 342–359. https://doi.org/10.1002/glia.23095.

Traka, M., Podojil, J. R., McCarthy, D. P., Miller, S. D., and Popko, B. (2016). Oligodendrocyte death results in immune-mediated CNS demyelination. *Nature Neuroscience*, 19(1), 65–74. https://doi.org/10.1038/nn.4193.